# Score-based 3D molecule generation with neural fields

**Matthieu Kirchmeyer**[*], **Pedro O. Pinheiro**[*], **Saeed Saremi**
Prescient Design, Genentech

## Abstract

We introduce a new representation for 3D molecules based on their continuous atomic density fields. Using this representation, we propose a new model based on walk-jump sampling [1] for unconditional 3D molecule generation in the continuous space using neural fields. Our model, FuncMol, encodes molecular fields into latent codes using a conditional neural field, samples noisy codes from a Gaussian-smoothed distribution with Langevin MCMC (walk), denoises these samples in a single step (jump), and finally decodes them into molecular fields. FuncMol performs all-atom generation of 3D molecules without assumptions on the molecular structure and scales well with the size of molecules, unlike most approaches. Our method achieves competitive results on drug-like molecules and easily scales to macro-cyclic peptides, with at least one order of magnitude faster sampling.[1]

## 1 Introduction

Generative modeling of 3D molecular structures, if deployed successfully, can help on many problems in material and life sciences. Recently, state-of-the-art image and text generative models were adapted to 3D molecule generation, achieving some degree of success [2, 3]. However, unlike other domains where the data modality is defined by the representation itself (e.g., a digital image *is* a tensor of pixels), there are multiple ways to represent a molecule. Therefore, an important problem to consider when modeling 3D molecules is: *what constitutes a good representation for molecules?*

Recent methods for 3D molecule generation usually represent molecules as point clouds of atoms [4] or discrete grids of atomic densities [5], which we will refer to as voxel grids. Point clouds are processed by graph neural networks (GNNs), usually based on equivariant architectures [6, 7]. GNNs are known to be less expressive than other architectures due to the message passing formalism [8, 9, 10] and often scale quadratically with the number of atoms. On the other hand, voxel grids are compatible with more expressive models (e.g., convnets and transformers) but computation and memory scales cubically with the volume occupied by the molecules. These limitations in expressivity and scalability hinder the scope of application of these models.

In this work, we propose a new representation for molecules that overcomes those limitations. Inspired by the 3D computer vision community [11], we represent *molecules as fields encoding atomic occupancy*, i.e., continuous functions that map 3D coordinates to atomic densities. Arguably this representation is more natural for molecules than for visual data: while visual data is obtained via discrete measurements, molecular fields are continuous by nature. We handle these fields as such, by parameterizing the molecular occupancy field with a neural network, shared among all molecules, and modulation codes, specific to each molecule. Fields that are parametrized by neural networks are referred to as neural fields, implicit neural representations (INR) or coordinate-based neural networks. The former models common molecular structures (e.g., bonds, angles, valencies, symmetries) while the later encodes variations that make each molecule unique. Given a modulation code, we decode the

---

[*] Equal contribution. Correspondence to `kirchmeyer.matthieu@gene.com`, `oliveira_pinheiro.pedro@gene.com`, `saremi.saeed@gene.com`
[1]The code is available at `https://github.com/prescient-design/funcmol`.

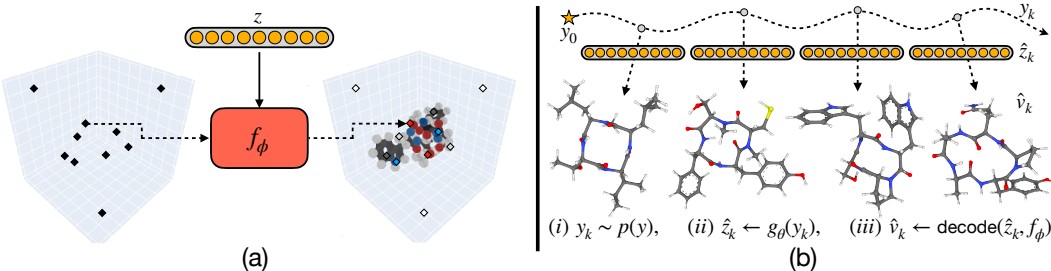

Figure 1: (a) a conditional neural field encodes a molecular field $v$ into a low dimensional latent code $z$. (b) using a learned score function $g_\theta$, FuncMol performs sampling in latent space via Langevin MCMC. These codes are decoded back into molecules.

molecular field by predicting the occupancy of each atom at given 3D coordinates (see Figure 1(a)). This decodes the molecules into explicit representations (such as discrete grids at arbitrary resolution or a `.sdf` format file), useful for downstream tasks.

We perform generative modeling in the continuous function space simply by sampling new modulation codes. Our proposed approach, *FuncMol*, leverages a modulation code denoiser to sample molecules following the (score-based) walk-jump sampling (WJS) approach [1]. WJS enjoys many properties such as fast-mixing, simplicity for training and fast sampling speed. Sampling is composed of three steps: (i) *(walk)* sample noisy modulation codes with a Langevin Markov chain Monte Carlo (MCMC), (ii) *(jump)* estimate the "clean" modulation codes, and (iii) *(decode)* convert the estimated codes into a molecule. Figure 1(b) illustrates a WJS chain, with samples generated by our model trained on a macrocyclic peptides dataset [12].

The neural molecular field representation has many advantages over prior representations: (i) it represents complex high-dimensional data in a relatively low-dimensional compact space, (ii) it is scalable (w.r.t. the number of points, size of molecules and resolution) and has low memory footprint, (iii) it does not make any assumptions on molecular structure or geometry, (iv) it can represent molecular structures at arbitrary resolutions and for a free-form discretization, (v) it is compatible with expressive machine learning architectures, and (vi) it is domain-agnostic and can be used for a variety of molecular design problems that can be expressed over fields, e.g., atomic densities, surfaces, pharmacophores, molecular orbitals, electron densities etc.

In summary, our contributions are as follows. We introduce a new way to represent molecular structures with neural fields. These representations are low-dimensional, compact, scalable and do not make any assumptions on the molecular structure. We then propose FuncMol, a score-based model for 3D molecule generation that leverages these representations. We show that FuncMol performs competitively against representative baselines on the drug-like molecules dataset GEOM-drugs [13], based on a wide set of standard and new metrics that we introduce to better measure the generation quality. These results were achieved with one order magnitude faster sampling time.[2] Finally, we illustrate FuncMol's ability to scale to larger 3D molecules by training it on CREMP [12], a recent macro-cyclic peptide dataset, to which our baselines are currently unable to scale.

## 2 Related work

**Neural fields.** Neural fields, also referred to as implicit neural representations (INRs), are coordinate-based neural networks that map coordinates (e.g., pixels on an image or coordinates in 3D Euclidean space) to features (e.g., RGB values or atomic occupancies). The idea of representing data points implicitly as neural networks dates back to the work of [14]. Recently, these representations have been successfully applied to model continuous signals, e.g., 2D images [15, 16, 17], 3D shapes [18, 19, 20, 21], 3D scenes [22, 23], videos [24, 25], physics [26, 27], due to their appealing properties. Recently, two concurrent seminal work lead to a fast progress of neural fields by overcoming the spectral bias of coordinate-based neural networks [28]. Sitzmann *et al.* [29] propose SIREN, a neural network that uses periodic activation functions, while Tancik *et al.* [30] considers a posi-

---

[2]Sampling time includes the "decoding" step to convert the generated code into a molecule.

tional encoding based on Fourier features. Built on top of those architectures, multiplicative filter networks (MFNs) [31] represent fields as a simple linear combination over an exponential number of basis functions (e.g. Fourier or Gabor basis). Due to their simplicity and strong performance, we use MFNs to model the atomic occupancy fields.

**Generative models of fields.** Generative models for neural fields were first applied in 3D computer vision problems. Mescheder *et al.* [19] learn the distribution of shape occupancy fields with VAEs [32], while [18, 33] achieves similar objectives using GANs [34]. Diffusion models [35] have also been applied to learn the distribution of neural fields [36, 37, 38, 39]. Some work [37, 40] parameterize the neural field with the vector of all the corresponding weights. However, when the signal is complex and the neural fields have large number of parameters (e.g., in the order of millions), it is preferable to parameterize the field with a latent code with much lower dimension [36, 41, 42, 43]. Dupont *et al.* [36] fit the whole dataset with a shared coordinate-based network and learn a latent modulation code for each field with gradient-based meta learning [44]. Similarly to them, we parameterize neural fields with latent modulation codes. However, instead of applying meta learning, we learn the latent codes through stochastic optimization, either following the "auto-encoding" [19] or the "auto-decoding" [20] framework.

**3D molecule generation.** Most 3D molecule generation approaches represent atoms as points (with coordinates and atom types) and molecules as a set of points. For example, [45, 46, 47] propose autoregressive approaches to sample atoms, while [48, 49] use normalizing flows [50]. Hoogeboom *et al.* [4] propose EDM, a diffusion model [35] applied to point cloud of atoms with E(3) equivariance [6]. Many follow-up works extend EDM [51, 52, 53]. For example, [54, 55, 56] improve its performance by leveraging extra information during training (such as molecular graph and formal charges). This contribution is orthogonal to ours and can potentially be incorporated into our generative model. Other approaches [57, 58] map atomic densities on discrete 3D regular grids and leverage computer vision techniques for generation. Recently, VoxMol [5] (and its latent version [59]), a score-based generative model based on walk-jump sampling [1], shows that voxel-based representations can achieve state-of-the-art results on 3D drug-like molecule generation. However, these methods scale cubically with the volume occupied by molecules, which limits its scope of application. Neural fields are the continuous generalization of discrete 3D grid representations: they achieve good performance on 3D molecule generation and are more efficient in terms of memory and computation.

**Conditional molecule generation.** Voxels and point-clouds have also been used for conditional 3D molecule generation, usually by building upon an unconditional model. The authors in [57] condition generation on 3D pharmacophores features, [60, 61, 62, 63] generate ligands conditioned on protein pockets, [64] generate molecules conditioned on fragments and [65, 66] generate 3D conformations conditioned on molecular graphs. We are aware of only one other work that uses field-based representation for molecules [67]. There are several differences between our works: they use different data representation, neural network architecture and noise model. While they consider the problem of generating molecule conformations given a molecular graph, we handle the more general problem of unconditional 3D molecule generation (without access to a molecular graph). Our model can easily be adapted to conformer generation by conditioning the generative model to the molecular graph. Moreover, our approach can also be conditioned to tasks where we do not have access to molecular graphs, such as structure-based drug design or electron density generation.

## 3 Neural atomic occupancy fields

We now describe how we represent molecules as continuous occupancy fields, how we approximate them with neural fields and how we decode the neural fields to retrieve molecular conformations. We finish the section by providing some useful properties of our neural field representations.

### 3.1 Molecules as continuous occupancy fields

We represent atoms as continuous Gaussian-like shapes in 3D space, centered around their atomic coordinates. Molecules are defined as fields mapping every point in the 3D space to the atomic densities of each atom type, $v : \mathbb{R}^3 \rightarrow \mathbb{R}^n$, where $n$ is the number of atom types in the dataset $\mathcal{D}$. We follow previous work [68, 69, 70], and compute the occupancy field $v_a$ for each atom type $a$ by

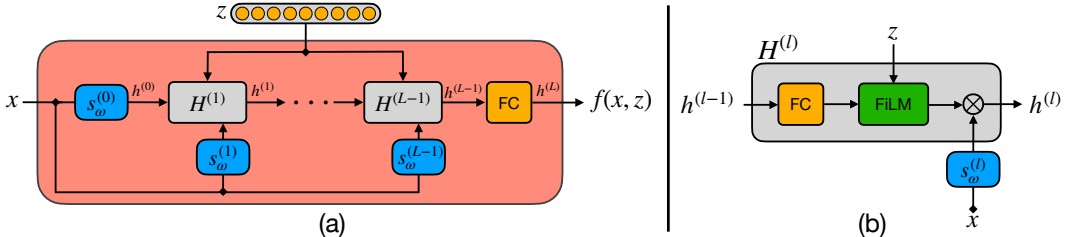

Figure 2: Conditional neural field $f_\phi$ using the multiplicative filter network architecture. (a) A latent code $z$ and some coordinates $x$ are given as input to the model that outputs the occupancy field at that location for the corresponding molecule, $f_\phi(x, z)$. (b) The code and coordinates are processed via FiLM layers and Hadamard products. We denote the overall operation at layer $l$ as $H^{(l)}$.

integrating the occupancy generated by all atoms of this type as follows:

$$\forall x \in \mathbb{R}^3, \ v_a(x) = 1 - \prod_{i=1}^{n_a} \left(1 - \exp\left(-\left(\frac{\|x - x_{a_i}\|}{.93r}\right)^2\right)\right), \tag{1}$$

where $a_i$ is the $i^{\text{th}}$ atom of type $a$, for a total of $n_a$ atoms. We set the atoms' radius to be $r = .5\text{Å}$ for all atom types. Molecular fields are smooth functions taking values between 0 (far away from all atoms) and 1 (at the center of atoms).

## 3.2 Molecular neural fields

Each molecule in the dataset is mapped to a modulation code $z \in \mathbb{R}^d$ and we parameterize the molecular occupancy $v$ with a conditional neural field $f_\phi : \mathbb{R}^3 \times \mathbb{R}^d \to \mathbb{R}^n$. Our objective is to learn the parameters $\phi$ and the modulation code $z$ such that for any molecular field $v$ and coordinate $x \in \mathbb{R}^3$, $f_\phi(x, z) \approx v(x)$. We approximate the molecular fields with a linear combination of an exponential large number of parameterized basis functions, such that amplitudes are modulated by the individual codes $z$. This parametrization is achieved by modeling the neural field with multiplicative filter networks (MFN) [31], a type of coordinate-based network that provides an elegant way to perform this linear combination under some assumptions on the basis functions. We introduce the parameters associated with these functions in Equation (4).

Our conditional MFN is a network with $L$ multiplicative blocks, as illustrated on Figure 2(a). We implement conditioning of its parameters with FiLM [71]. Each multiplicative block is composed of a fully-connected layer, a FiLM modulation layer and an elementwise product with a basis, as illustrated on Figure 2(b). The neural field can be expressed by the following recursive expression:

$$h^{(0)}(x) = s_{\omega^{(0)}}(x),$$
$$h^{(l)}(x) = \left(\gamma^{(l-1)} \odot \left(W_f^{(l-1)} h^{(l-1)}(x)\right) + \left(b^{(l-1)} + \beta^{(l-1)}\right)\right) \odot s_{\omega^{(l)}}(x), \ l \in (1, L-1),$$
$$f_\phi(x, z) \triangleq h^{(L)}(x) = W_f^{(L-1)} h^{(L-1)}(x) + b^{(L-1)},$$

where $s_{\omega^{(l)}}$ is a spatial basis function parameterized by $\omega^{(l)}$, $\odot$ denotes the Hadamard product and

$$\beta^{(l-1)} = W_\beta^{(l-1)} z, \ \text{and} \ \gamma^{(l-1)} = W_\gamma^{(l-1)} z,$$

are the bias and scale modulation terms. We propose two approaches to learn the neural field's parameters $\phi = \{W_f^{(l)}, b^{(l)}, \omega^{(l)}, W_\beta^{(l)}, W_\gamma^{(l)}\}$ and the codes $z$ (one per each molecule in $\mathcal{D}$).

**Auto-decoding.** In this setting, introduced by [20], we initialize each code $z$ randomly and directly learn them (together with the parameters of the neural field) with backpropagation. This is achieved by solving the following optimization problem:

$$\underset{\phi, \{z_v\}_{v \in \mathcal{D}}}{\arg\min} \sum_{v \in \mathcal{D}} \int \|f_\phi(x, z_v) - v(x)\|_2^2 \, dx, \tag{2}$$

where the integral is approximated by sampling finite sets of points $\mathcal{X} \subset \mathbb{R}^3$. While auto-decoding was usually applied in settings with relatively few samples, we were able to scale the training to large datasets of one million samples (see Appendix B). See Algorithm 1 on Appendix B for more details.

**Auto-encoding.** This approach, introduced by [19] and illustrated in Appendix B Figure 4, generates the modulation code via an encoder $\zeta_\psi$, parameterized by $\psi$ and decodes the neural field back. In this work, $\zeta_\psi$ is a (trainable) 3D convolutional network encoder that takes (low-resolution) voxel grids $\mathcal{G}$ as inputs. This approach is flexible and compatible with other encoder architectures and molecule representations (e.g., GNN/point clouds). The parameters of the encoder and the neural field are learned with the following objective:

$$\arg\min_{\phi,\psi} \sum_{v \in \mathcal{D}} \int \|f_\phi(x, \zeta_\psi(\mathcal{G})) - v(x)\|_2^2 \, \mathrm{d}x. \tag{3}$$

Once training is done, we generate the code with the trained encoder. See Algorithm 2 on Appendix B for more details. Instead of learning the codes individually, this approach learns an encoder, which allows to leverage data augmentation more efficiently. As a result it helps learn a more structured latent space. These benefits are reflected empirically in our experiments.

### 3.3 From codes to atomic coordinates

By leveraging the modulation codes $z$ and the neural field $f_\phi$, we have access to the (learned) continuous occupancy field, $f_\phi(\cdot, z)$. However, in many useful applications in chemistry and biology, we are more interested in the 3D conformation of molecules. Next, we describe how we can extract the molecular conformation from a learned (or generated, as we will see next) modulation code. This is the decoding step outlined in the introduction.

We start by identifying all atoms in the field, their approximate locations and their type. To this end, we render a discretized voxel grid from the molecular field using a uniform discretization of space and the neural field $f_\phi(\cdot, z)$. We then apply a *peak finding* algorithm to infer the number of atoms in the molecule on each channel of the grid (each representing a separate atom type) and their (discretized) coordinates. Finally, we introduce a new continuous refinement to find the local maximum of the neural field. For each identified atoms $a$, we refine its coordinates around the neighborhood of the coordinates found with the peak detector $x_a^0$:

$$x_a = \arg\max_{x \in \mathbb{R}^3 : \|x - x_a^0\| \leq r} [f_\phi(x, z)]_a,$$

where $[f_\phi(x, z)]_a$ denotes the field restricted to the channel corresponding to the atom type. This continuous refinement finds atomic coordinates that lie beyond the initial coarse uniform discretization. In practice, we batch the refinement process across molecules and use L-BFGS. We demonstrate in Appendix E.2 its efficiency compared to prior non-continuous refinement approaches from [72, 5].

### 3.4 Molecular neural fields properties

The proposed conditional neural field enjoys many properties that make it a natural choice for handling large 3D molecules represented as continuous fields.

**Flexibility w.r.t. basis.** Conditioning MFNs gives the flexibility to choose any type of spatial basis that satisfies a multiplicative-sum property (see the definition in [31]). In our preliminary experiments, we observed that setting the spatial basis to Gabor filters performed better than Fourier filters as they account for the sparse nature of occupancy fields. For each layer $l$, we consider the following Gabor parameterization, also used in [27]:

$$s_{\omega^{(l)}}(x) = \exp\left(-\frac{\nu^{(l)}}{2}\|x - \mu^{(l)}\|_2^2\right)(\cos(\Omega^{(l)}x), \sin(\Omega^{(l)}x)), \tag{4}$$

where $\mu^{(l)}$ is the mean of the Gabor filter, $\nu^{(l)}$ is the scale, $\Omega^{(l)}$ is the frequency and $(\cdot, \cdot)$ refers to the concatenation operator. Equation (4) combines both real and imaginary parts of the complex Gabor filter. This allows to remove phase parameters and reduce the overall parameter count of MFNs [31]. Other choices of basis are also possible and are left for future work.

**Parameter efficiency.** Our overall conditional MFN formulation is parameter efficient and shares parameters across molecules and channels (i.e. atom types). As [27], we excluded the basis functions parameters $\omega$ from FiLM to further decrease the parameter count.

**Memory efficiency.**   Our conditional neural field can be trained on any free-form discretization of the input field. Occupancy values are computed on the fly. This allowed to train FuncMol with large batch size even on large 3D molecules. We found that training the neural field by up-sampling points close to the atoms' center improved training time as further detailed in Appendix B. Alternative approaches like VoxMol [5] cannot be trained efficiently in this setting: for reference, on the macro-cyclic peptide generation task of Section 5.4, on 4 A100 GPUs VoxMol's training cost per epoch was 10 hours while our neural field's training cost was less than 12 minutes.

**Reconstruction quality and robustness to noise.**   Finally our neural field reconstructs accurately the input data as demonstrated in Appendix E.1. Moreover, operating on these latent codes makes our model extremely robust to noise in code space. We demonstrate this property in Appendix E.3 by reporting the sampling metrics when perturbing the codes $z$ by a Gaussian noise.

**Sampling efficiency.**   We use the latent codes for generative modeling as explained in Section 4. Most sampling operations are done on a small dimensional latent space, while decoding into a full molecular field is done only after sampling. As we show in Section 5, our approach (which involves sampling latent code followed decoding them into molecules) achieves at least one order magnitude faster molecule sampling time than previous methods.

## 4   Score-based generative modeling

We use our latent modulation representations for a downstream generative modeling task. Section 4.1 describes the neural empirical Bayes (NEB) formalism used in our method and Section 4.2 explains how we perform sampling.

### 4.1   Neural empirical Bayes

Let $p(z)$ be the distribution of codes and $p(v)$ be the (unknown) distribution of molecular fields, defined more formally as the pushforward of $p(z)$ via the mapping $z \mapsto f_\phi(\cdot, z)$. NEB estimates the score function of a smoothed density of the codes $p(y)$, $g_\theta(y) \approx \nabla \log p(y)$. Indeed sampling from a smoothed density $p(y)$ benefits from faster mixing than on the original density $p(z)$ [1, 73, 74]. This smoothed distribution is defined by transforming the random variable $Z$ with an additive isotropic Gaussian noise with a known noise level $\sigma$, $Y = Z + N$, where $N \sim \mathcal{N}(0, \sigma^2 I_d)$. The noise level $\sigma$ plays a key role, trading-off simplicity of the denoising objective and the sampling quality.

NEB is based on an empirical Bayes view of (denoising) score-based models that relates the estimator of clean data (denoiser) and the score function of the smoothed density at a fixed noise level [75, 76, 1]. The denoiser is taken to be the least-square estimator of $Z$ given $Y = y$ which is the Bayes estimator, i.e. $\hat{z}(y) = \mathbb{E}[Z|Y = y]$. Under Gaussian noise, denoiser and smoothed score function are related by the Tweedie-Miyasawa formula:

$$\hat{z}(y) = y + \sigma^2 \nabla \log p(y). \tag{5}$$

The denoiser is parameterized by a neural network and learned by minimizing the following objective:

$$\mathcal{L}(\theta) = \mathbb{E}_{z \sim p(z), \varepsilon \sim \mathcal{N}(0, \sigma^2 I_d)} \big\| z - \hat{z}_\theta(z + \varepsilon) \big\|_2^2. \tag{6}$$

The score function is recovered from a learned denoiser via Equation (5) and is used for sampling smoothed codes (see Section 4.2). In practice, we optimize the empirical loss based on the latent codes inferred from a set of molecular fields $\mathcal{D}$. See pseudo-code in Appendix B, Algorithm 3.

### 4.2   Walk-jump sampling

We use the score function $g_\theta$ to sample codes using the *walk-jump sampling* (WJS) scheme [1, 77, 73, 78]. This approach samples molecules from $p(z)$ using the learned score function of noisy data instead of clean data. It consists of two main steps: walking and jumping as detailed in Appendix B, Algorithm 4. Figure 1(b) illustrates these two main steps in a WJS chain: walking consists in generating noisy codes while jumping consists in generating clean codes $z$.

*(initialization)* To improve mixing, as [77], we initialize the chains by adding uniform noise to Gaussian noise (with the same $\sigma$ used when training the denoiser). In practice we sample the uniform

noise over the range of code values, $\mu \sim \mathcal{U}_d(\min_{z \in \mathcal{D}_z, i \in \{1 \cdots d\}} z_i, \max_{z \in \mathcal{D}_z, i \in \{1 \cdots d\}} z_i)$, where $\mathcal{D}_z$ is the training dataset of codes, arriving at $y_0 = \mu + \varepsilon, \varepsilon \sim \mathcal{N}(0, \sigma^2 I_d)$.

*(walk)* Noisy codes are sampled from $p(y)$ with Langevin MCMC algorithms that discretize the underdamped Langevin diffusion [79] starting from $y_0$ and $u_0 = 0$:

$$\mathrm{d}u_t = -\gamma u_t \mathrm{d}t \ + \ g_\theta(y_t)\mathrm{d}t \ + \ \sqrt{2\gamma}\,\mathrm{d}B_t, \qquad \mathrm{d}y_t = u_t \mathrm{d}t \ , \qquad (7)$$

where $B_t$ is the standard Brownian motion in $\mathbb{R}^d$ and $\gamma$ is the friction (the "mass" is set to 1). We discretize this SDE using the ABOBA scheme from Sachs *et al.* [80], given a discretization step $\delta$ and a fixed number of walk steps $K$. We analyze the impact of $K$ in Appendix E.4.

*(jump)* At a given time step $K$, clean samples are estimated by denoising the smooth code, i.e., $z_K = \hat{z}_\theta(y_K)$. These codes are then used to obtain the atomic coordinates as detailed in Section 3.3.

## 5 Experiments

We now evaluate our model for unconditional generation. We start with a description of our experimental setup (Section 5.1), then present our results on two popular small molecule datasets (Sections 5.2 and 5.3) and a recent macro-cyclic peptide dataset (Section 5.4).

### 5.1 Experimental setup

**Datasets.** We evaluate FuncMol on three datasets: *QM9* [81], *GEOM-drugs* [82] and *CREMP* [12]. QM9 contains an enumeration of all possible molecules up to 9 heavy atoms (29 including hydrogens) satisfying some constraints [83]. GEOM-drugs contains multiple conformations for 430K drug-sized molecules (computed with semi-empirical density functional theory), with an average of 44 heavy atoms per molecule. CREMP is a recent dataset that contains multiple conformations of macrocyclic peptides 4-6 residue long, with an average of 74 heavy atoms per molecule. We model hydrogen explicitly and consider 5 chemical elements for QM9 (C, H, O, N, F), 6 for CREMP (C, H, O, N, F, S) and 8 for GEOM-drugs (C, H, O, N, F, S, Cl and Br), ignoring the P, I and B elements that occur extremely rarely. We use a split of 100K/20K/13K molecules for QM9, 1.1M/146K/146K on GEOM-drugs and 409K/10K/9K on CREMP for train, validation and test, respectively. We use the same pre-processing and splits in [54] for QM9 and GEOM-drugs and in [84] for CREMP.

**Implementation details.** Our main model, *FuncMol*, follows the auto-encoding approach described in Section 3.2. The codes $z$ are computed with an encoder that takes as input a low-resolution voxelized representation of the molecular field with grid dimension of $16 \times 16 \times 16$. The encoder is a 3D CNN containing 4 residual blocks, where each block contains 3 convolutional layers followed by BatchNorm, ReLU and pooling (max pooling on the first three blocks and average pooling on the last one) layers. We consider modulation codes with dimension 1024 on QM9 and 2048 on GEOM-drugs and CREMP. We use the same neural field network for all datasets: a conditional MFN with Gabor filters and 6 FiLM-modulated layers, where each fully-connected layer has 2048 hidden units. We augment the training set by applying random rotations on the three Euler angles. The weights of the latent code encoder and neural field decoder are trained jointly.

We also show results for the auto-decoding based model, *FuncMol*dec. In this setting, we initialize the codes randomly and optimize them together with the neural field weights. This approach is less fit for performing large amounts of augmentation as it solves a costly per-sample optimization problem; thus we did not apply data augmentation. As a consequence, we observed that this model is more prone to memorization than the auto-encoding approach (e.g., on GEOM-drugs, around 33% of the generated molecules are copies from the training set).

We normalize the codes to have zero mean and unit variance. We choose a noise level in normalized space of $\sigma = 1.2$ for GEOM-drugs and CREMP, $\sigma = 2.0$ for QM9. Our code denoiser is a modified version of the denoiser used in [36]: a fully-connected network with 18 residual blocks (each with two linear layers with 6144 hidden units) and skip connections. We remove the bias of all layers and use ReLU activations as in [85]. To limit memorization in FuncMoldec, we add dropout (ratio 0.3) between the fully-connected layers in each residual block. For QM9, we consider a smaller network (6 residual blocks and 4096 hidden units). We initialize the MCMC chains with noise and use the following sampling hyperparameters $\gamma = 1.0$ and $\delta = \sigma/2$ as in [5, 78]. For evaluation purposes, we

generate one sample per chain. We consider 1000 steps for QM9 and GEOM-drugs and 10000 steps for CREMP. See Appendix B for more details on the implementation.

**Baselines.** We compare FuncMol and FuncMol$_{\text{dec}}$ to three state-of-the-art approaches. *EDM* [4] and *GeoLDM* [53] are diffusion models operating on point clouds (the latter is a latent-space extension of the former). *VoxMol* [5] is a voxel-based generative model that uses neural empirical Bayes, similar to our generative approach. All of the methods generate molecules as a set of atom types and their coordinates. EDM and GeoLDM apply diffusion directly to point clouds, while VoxMol and FuncMol rely on an additional (cheap) post-processing step to extract atomic coordinates from voxel grids or modulation codes, respectively. We follow previous work [58, 54, 5, 62, 86], and use standard cheminformatics software (OpenBabel [87]) to determine the molecule's atomic bonds given their atomic coordinates. The same post-processing is applied to all models for fairness of comparison.

**Metrics**. We consider several metrics used in previous work [5] to benchmark unconditional molecule generation for the standard QM9 and GEOM-drugs datasets (for the CREMP metrics, see Section 5.4): *stable mol* and *stable atom*, the percentage of stable molecules and atoms (as defined in [4]); *validity*, the percentage of generated molecules that passes RDKit [88]'s sanitization filter; *uniqueness*, the proportion of valid molecules that have different canonical SMILES; *valency* $W_1$, the Wasserstein distance between the distribution of valencies in the generated and test set; *atoms TV* and *bonds TV*, the total variation between the distribution of atom types and bond types; *bond length* $W_1$ and *bond angle* $W_1$, the Wasserstein distance between the distribution of bond and lengths. We also report the *average sampling time per molecule*. In the case of our method, this time includes the MCMC "walk" steps, the denoising "jump", the rendering, peak detection and bond inference.

To further investigate the quality of molecular conformations and other molecular properties on GEOM-drugs, we consider some additional metrics. These include: *single fragment*, the percentage of molecules that contains only a single fragment; *median strain energy* [89], the difference between the internal energy of the generated molecule's pose and a relaxed pose of the molecule using RDKit's Universal Force Field [90], computed over all molecules; *ring size TV*, the total variation between the empirical distribution of ring sizes (i.e. number of heavy atoms in rings) in generated and test sets; *number of atoms/mol TV*, the total variation between the empirical distribution of number of atoms per molecule in generated and test sets (in the case of molecules with multiple fragments, we consider only the largest fragment); *QED, SA and logp*, measure the drug-likeness score [91], the synthesizability score [92] and the lipophilic efficiency, respectively (computed with RDKit).

**Ablations.** In Appendix E we report a series of ablation studies for the neural field and the generative model. Appendix E.1 measures the reconstruction quality of the training molecules. Appendix E.2 illustrates the improvements due to continuous atomic coordinate refinement. Appendix E.3 shows that our field-based decoder is robust to noise, making it an ideal choice for generative modeling. Appendix E.4 ablates the impact of the number of walk steps in the WJS scheme of Section 4.2. Finally, Appendix E.5 ablates the impact of the chosen resolution when sampling codes and decoding them back to molecules. In practice, we observe that 0.25Å provides a good trade-off between the sampling time and the quality of the generated molecules.

## 5.2   Results on QM9

As pointed by previous authors [93, 4], this dataset is not fully suited for unconditional generative models: a model that captures the training distribution will have to generate samples from training set, due to the enumeration. However, many previous work report results on this dataset. Therefore, we also show results for completeness.

Table 1 report the metrics described in Section 5.1. We see that FuncMol slightly improves VoxMol and both models perform worse compared to the equivariant point-cloud based baselines. We note that sampling time of FuncMol is an order of magnitude better than baselines.

## 5.3   Results on GEOM-drugs

Table 2 reports the same set of metrics as in the previous dataset. FuncMol performs favorably over point cloud diffusion models and is close to VoxMol's performance. In particular, FuncMol and VoxMol generate molecules that are significantly more stable and better capture the distribution of bond angles. Table 3 shows results on additional metrics (described in Section 5.1). We also include

Table 1: QM9 results w.r.t. test set for 10000 samples per model. ↑/↓ indicate that higher/lower numbers are better. The row *data* are randomly sampled molecules from the validation set. We report 1-sigma error bars over 3 sampling runs.

| | stable mol %↑ | stable atom%↑ | valid %↑ | unique %↑ | valency $W_{1↓}$ | atom $TV_↓$ | bond $TV_↓$ | bond len $W_{1↓}$ | bond ang $W_{1↓}$ | time s/mol.↓ |
|---|---|---|---|---|---|---|---|---|---|---|
| *data* | 98.7 | 99.8 | 98.9 | 99.9 | .001 | .003 | .000 | .000 | .120 | - |
| EDM | 97.9 | 99.8 | 99.0 | 98.5 | .011 | .021 | .002 | .001 | .440 | 0.54 |
| GeoLDM | 97.5 | 99.9 | 100. | 98.0 | .005 | .017 | .003 | .007 | .435 | 0.65 |
| VoxMol | 89.3 | 99.2 | 98.7 | 92.1 | .023 | .029 | .009 | .003 | 1.96 | 0.83 |
| FuncMol$_{dec}$ | 88.6 | 99.2 | 100. | 81.1 | .022 | .066 | .032 | .006 | 1.21 | 0.05 |
| FuncMol | 89.2 (±.4) | 99.0 (±.07) | 100. (±0) | 92.8 (±0.3) | .021 (±0.001) | .012 (±0.001) | .006 (±0.003) | .005 (±0.009) | 1.56 (±0.06) | 0.05 |

Table 2: GEOM-drugs results, standard metrics w.r.t. test set for 10000 samples per model. ↑/↓ indicate that higher/lower numbers are better. The row *data* are randomly sampled molecules from the validation set. We report 1-sigma error bars over 3 sampling runs.

| | stable mol %↑ | stable atom%↑ | valid %↑ | unique %↑ | valency $W_{1↓}$ | atom $TV_↓$ | bond $TV_↓$ | bond len $W_{1↓}$ | bond ang $W_{1↓}$ | time s/mol.↓ |
|---|---|---|---|---|---|---|---|---|---|---|
| *data* | 99.9 | 99.9 | 99.8 | 100.0 | .001 | .001 | .025 | .000 | 0.05 | - |
| EDM | 40.3 | 97.8 | 87.8 | 99.9 | .285 | .212 | .048 | .002 | 6.42 | 9.35 |
| GeoLDM | 57.9 | 98.7 | 100. | 100. | .197 | .099 | .024 | .009 | 2.96 | 8.96 |
| VoxMol | 75.0 | 98.1 | 93.4 | 99.6 | .254 | .033 | .024 | .002 | 0.64 | 7.55 |
| FuncMol$_{dec}$ | 69.7 (±.6) | 95.3 (±.1) | 100. (±.0) | 77.5 (±.6) | .268 (±.001) | .035 (±.001) | .028 (±.001) | .003 (±.000) | 2.13 (±.01) | 0.29 |
| FuncMol | 69.7 (±.2) | 98.8 (±.0) | 100. (±.0) | 95.3 (±.1) | .245 (±.001) | .109 (±.001) | .052 (±.000) | .003 (±.000) | 2.49 (±.06) | 0.29 |

the following plots of Appendix F: Figure 9 shows the cumulative distribution function of strain energies for generated molecules and Figures 10 and 11 show the histograms of the other metrics.

The results are clear: *FuncMol samples better drug-like molecules than point-cloud diffusion models*. In fact, about half the molecules of point cloud methods have multiple fragments, they have an order of magnitude higher median strain energy, the distribution of ring sizes is off and the QED, SA and logp scores are lower. The results of FuncMol are close to VoxMol in most but not all metrics. However, our approach is much more scalable and efficient: *FuncMol generates molecules an order of magnitude faster than previous methods* (see the last column of Table 2). Appendix H shows some molecules generated by FuncMol on GEOM-drugs.

Table 3: GEOM-drugs results, additional metrics w.r.t. test set for 10000 samples per model. ↑/↓ indicate that higher/lower numbers are better. The row *data* are randomly sampled molecules from the validation set. We report 1-sigma error bars over 3 sampling runs.

| | single frag %↑ | median energy↓ | ring sz $TV_↓$ | atms/mol $TV_↓$ | QED ↑ | SA ↑ | logp ↑ |
|---|---|---|---|---|---|---|---|
| *data* | 100. | 54.5 | .011 | .000 | .658 | .832 | 2.95 |
| EDM | 42.2 | 951.3 | .976 | .604 | .472 | .514 | 1.11 |
| GeoLDM | 51.6 | 461.5 | .644 | .469 | .497 | .593 | 1.05 |
| VoxMol | 82.6 | 69.2 | .264 | .636 | .659 | .762 | 2.73 |
| FuncMol$_{dec}$ | 80.2 (±.6) | 96.4 (±1.1) | .324 (±.008) | .970 (±.008) | .677 (±.015) | .788 (±.038) | 2.87 (±.00) |
| FuncMol | 70.5 (±.2) | 109.7 (±1.1) | .427 (±.006) | 1.05 (±.00) | .713 (±.001) | .811 (±.005) | 3.09 (±.02) |

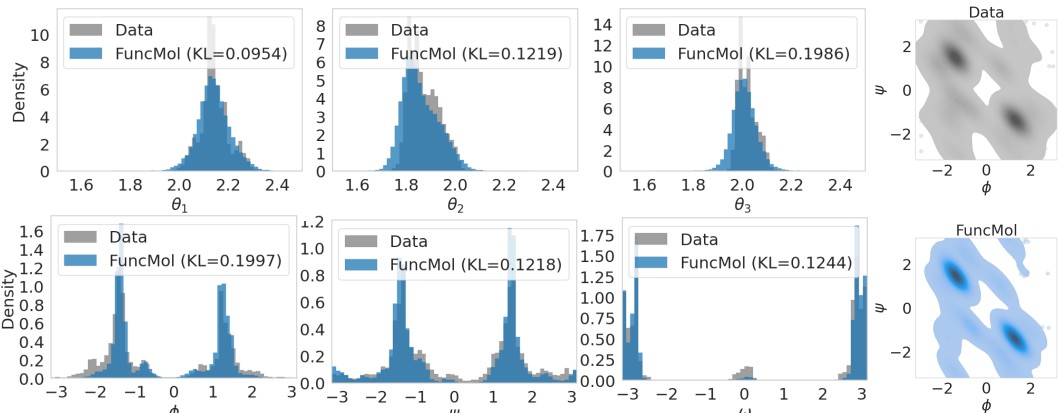

Figure 3: Qualitative evaluation on CREMP following [84]. Left: Comparison of the bond angles ($\theta_1$, $\theta_2$, $\theta_3$) in each residue and dihedral distributions ($\phi$, $\psi$, $\omega$) for each residue from the reference test set (gray) and the generated samples (blue). KL divergence is calculated as KL(test || sampled). Right: Ramachandran plots [94] (colored by density where darker tones represent high density regions).

## 5.4  Results on CREMP

To showcase the scalability of FuncMol, we train it on a dataset of larger molecules. We choose the macrocylic peptides of CREMP, that contains on average 74 atoms, making it challenging to train models using point-clouds. These molecules also pose serious limitations to voxel-based approaches as they require modeling a volume of $24^3$ cubic Angstroms. We tried to train VoxMol on this dataset using the official implementation, but did not succeed: it takes around 10 hours per epoch on 4 A100 GPUs, while FuncMol takes less than 12 minutes per neural field epoch and 15s per denoiser epoch. We use the same code dimension and neural field architecture as in GEOM-drugs, therefore the computational training cost of FuncMol remains unchanged.

Figure 3 shows that FuncMol captures well the underlying distribution of macrocyclic conformations. We show the distribution of bond angles ($\theta_1$, $\theta_2$, $\theta_3$) and dihedrals ($\phi$, $\psi$, $\omega$) of both molecules from test set and generated molecules. We also show the KL-divergence between test and generated samples. Approximately 65% of the generated molecules were valid peptides (that is, we could extract a sequence of amino acids from the SMILES strings). The Ramachandran plots [94] show that FuncMol recovers the main modes of the distribution. We note that the bond angles and dihedrals distributions are learned without having any explicit priors on the structure of these peptides. Appendix H shows some generated macrocylic peptides. Finally, our model takes around 1.5s to generate a molecule. For reference, should VoxMol be trained successfully, it would take over a minute to sample a single molecule (assuming similar sampling parameters as in other datasets). This is a substantial speedup that showcases the potential of FuncMol to scale to even larger molecules.

## 6  Discussion

We introduce a new continuous representation of 3D molecules based on their atomic occupancy field and a score-based generative model operating on this representation. Each molecule is assigned a code that modulates a shared neural field. We demonstrate that we can build an all-atom generative model of 3D molecules, FuncMol, with state-of-the-art sampling time and competitive performance on challenging drug-like datasets. We believe that this model introduces a new paradigm for all-atom 3D modeling of molecules that has many useful properties, namely scalability, expressivity, and flexibility, as it can model various molecular design problems (involving structure, electron densities, etc.) with minor architecture changes. Future research directions include exploring different neural field architectures, adapt the model for conditional generation (e.g., structure conditioning) or model the molecular bonds alongside the atomic coordinates[3]. Moreover, the scalability of FuncMol can be a potential alternative for all-atom representations of large biomolecules.

---

[3]Recent work [54, 95] show that this improves generation quality. See Appendix G to see how our method compares with a representative baseline that uses the bond information.

**Acknowledgements**    We would like to thank the Prescient Design team for helpful discussions and Genentech's HPC team for providing a reliable environment to train and analyze models.

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

# Appendices

This supplementary material is organized as follows:

## A   Broader Impact Statement

This work introduces some technical advancements in unconditional 3D molecule generation, an important component of molecular design and pharmaceutical research. A key advantage of our model is that it scales to larger molecules unlike existing models and has at least one order magnitude faster sampling time. Although extensive validation through wet-lab experiments and clinical trials is necessary, successful developments in this area have the potential to enhance human health, impacting a wide number of fields such as drug discovery, biology, materials science to cite a few. As with any technology, ensuring safe, ethical, and accountable deployment of these models is necessary to guarantee a positive impact on society.

## B   Implementation details

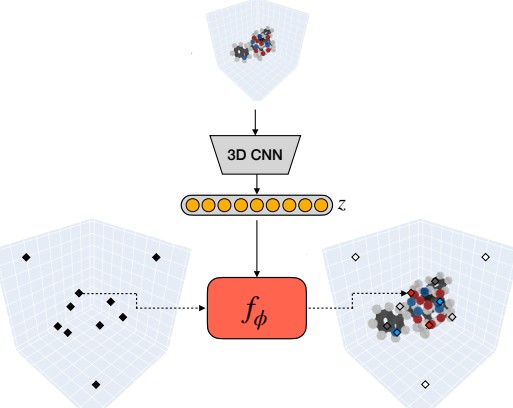

Figure 4: Auto-encoding approach for neural field representation. A voxelized representation of molecule is encoded int the latent space $z$ with a 3D CNN. This representation is then decoded with a conditional MFN for any point $x$ in space.

Here, we provide some more implementation details that complement Section 5.

**Conditional neural field.** The codes $z$ are computed with an encoder that takes as input a low-resolution voxelized representation of the molecular field with grid dimension of $N \times 16^3$. We use $N = 5$ for QM9, $N = 8$ for GEOM-drugs and $N = 6$ for CREMP. We use resolution of .5Å to generate the low-resolution grid on QM9, 1Å on GEOM-drugs and 1.667Å for CREMP. Before

voxelizing the molecules, we first center the atoms around the tightest bounding box encapsulating the molecule, apply a random rotation to the atoms (each Euler angle rotated randomly between $[0,2\pi)$) and normalize their coordinates to the range of $[-1, 1]$. The encoder is a 3D CNN containing 4 residual blocks (number of hidden units 256, 512, 1024, 2048 for each block), where each block contains 3 convolutional layers followed by BatchNorm, ReLU and pooling layers (we use max pooling on the first three blocks and average pooling on the last one). The encoder has 145M on QM9 and 229M on GEOM-drugs and CREMP. In the case of FuncMol$_{\text{dec}}$, we do not use any encoder and directly optimize the codes $z$, one for each molecule in the dataset.

The neural field and codes are optimized over a free-form discretization grid $\mathcal{X}$, that changes at each iteration. For each training step, we sample a random training molecule and randomly pick $N = 4000$ points, half of the points are taken out of an uniform discretization grid of resolution .25Å, and the remaining points are sampled equally across cubes of size $3\times3\times3$ and resolution .25Å, centered on each atom in the molecule. We found that this choice helped speed up training. For each point, we compute the atomic occupancy value for each atom using Equation (1).

The parameters of the conditional neural field are optimized with Adam. For FuncMol, we use a learning rate of $10^{-4}$ for the encoder and $5 \times 10^{-4}$ for the decoder using a node of 2 A100 GPUs with a batch size of 96 per GPU. For FuncMol$_{\text{dec}}$, we efficiently scaled auto-decoding to large datasets by optimizing the codes with SparseAdam, using a learning rate $10^{-3}$. The decoder optimizer is Adam with a learning rate of $10^{-3}$. We train the models for 900 epochs on QM9, 300 epochs on GEOM-drugs and 1000 epochs on CREMP. Algorithm 1 and Algorithm 2 provide pseudocodes for learning the conditional neural field decoder and the latent codes (FuncMol$_{\text{dec}}$) or the encoder (FuncMol).

---

**Algorithm 1:** Auto-decoding conditional neural field training pseudo-code—Equation (2)

---

**Input :** $\mathcal{D}$ dataset of molecular fields, $\{z_v \leftarrow 0\}_{v\in\mathcal{D}}$ codes, $\phi \leftarrow \phi_0$ conditional MFN
  parameters; $N$ number of points to sample

**while** *not converged* **do**
  **for** *batch* $\mathcal{B} \subset \mathcal{D}$ **do**
    Sample a discretization grid $\mathcal{X}$ and compute occupancy $v(x), \forall v \in \mathcal{B}, \forall x \in \mathcal{X}$
    $\ell_{\text{dec}}(\phi, \{z_v\}_{v\in\mathcal{B}}, \mathcal{X}) = \sum_{v\in\mathcal{B}, x\in\mathcal{X}} \|f_\phi(x, z_v) - v(x)\|_2^2$
    $\{z_v\}_{v\in\mathcal{B}} \leftarrow \{z_v\}_{v\in\mathcal{B}} - \eta_z \nabla_z \ell_{\text{dec}}(\phi, \{z_v\}_{v\in\mathcal{B}}, \mathcal{X});$       `/* Update codes */`
    $\phi \leftarrow \phi - \eta_\phi \nabla_\phi \ell_{\text{dec}}(\phi, \{z_v\}_{v\in\mathcal{B}}, \mathcal{X});$       `/* Update decoder weights */`

---

**Algorithm 2:** Auto-encoding conditional neural field training pseudo-code—Equation (3)

---

**Input :** $\mathcal{D}$ dataset of molecular fields, $\psi \leftarrow \psi_0$ voxel encoder parameters; $\phi \leftarrow \phi_0$ conditional
  MFN parameters; $N$ number of points to sample, uniform "low-resolution" voxel grid $\mathcal{G}$

**while** *not converged* **do**
  **for** *batch* $\mathcal{B} \subset \mathcal{D}$ **do**
    Sample a discretization grid $\mathcal{X}$ and compute occupancy $v(x), \forall v \in \mathcal{B}, \forall x \in \mathcal{X}$ and
    low-resolution voxel grid $\mathcal{G}_v, \forall v \in \mathcal{B}$.
    $\ell_{\text{dec}}(\phi, \psi, \mathcal{X}, \mathcal{B}) = \sum_{v\in\mathcal{B}, x\in\mathcal{X}} \|f_\phi(x, \zeta_\psi(\mathcal{G}_v)) - v(x)\|_2^2$
    $\phi \leftarrow \phi - \eta_\phi \nabla_\phi \ell_{\text{dec}}(\phi, \psi, \mathcal{X}, \mathcal{B});$       `/* Update decoder weights */`
    $\psi \leftarrow \psi - \eta_\psi \nabla_\psi \ell_{\text{dec}}(\phi, \psi, \mathcal{X}, \mathcal{B});$       `/* Update encoder weights */`

---

**Modulation code denoiser** $\hat{z}_\theta$**.** Once the modulation codes and the conditional neural field are learned, we pre-process the codes to have zero mean and unit variance, then learn a denoiser in normalized space using $\sigma = 1.2$ on GEOM-drugs and CREMP and $\sigma = 2$ for QM9, following Algorithm 3.

Our denoiser has a projection linear layer (that embed the 1024 / 2048 code into a 6144 space) followed by several residual blocks, where each block contains (in this order): group normalization layer, ReLU non-linearity, fully-connected layer, normalization layer, ReLU non-linearity, drop-out with rate 0.3 for FuncMol$_{\text{dec}}$ or none for FuncMol and another fully-connected layer. We then add one final layer to go back to the original 1024 / 2048 code space. We use similar "skip-connections" as in the MLP denoiser of [33], adapted from 2D U-Net architectures [96]. For GEOM-drugs and CREMP, we consider a model with 1.9B parameters (12 residual blocks, 6144 hidden units). For QM9, we train a model of size 445M parameters (6 residual blocks, 4096 hidden units). The models

are trained with batch size 2048 on a single A100 GPU for 2500 epochs with AdamW [97] (learning rate $10^{-3}$, weight decay $10^{-2}$) and exponential moving average (EMA) with a decay of .9999. As [36], we use the following learning rate schedule: we warm-up the learning rate linearly from 0 to 3e-4 for the first 4000 iterations, then decay it proportionally to the square root of the iteration count. The pseudo-code is given in Algorithm 3.

---

**Algorithm 3:** Denoiser training pseudo-code - Equation (6)

**Input :** $\mathcal{D}_z = \{z_v\}_{v \in \mathcal{D}}$ normalized codes, denoiser $\hat{z}_\theta$
**while** not converged **do**
    **for** batch $\mathcal{B} \subset \mathcal{D}_z$ **do**
        $y \leftarrow z + \varepsilon, \quad \varepsilon \sim \mathcal{N}(0, \sigma^2 I_d)$
        $\ell_{\text{denoiser}}(\theta, \mathcal{B}) = \sum_{z \in \mathcal{B}} \|z - \hat{z}_\theta(y)\|_2^2$
        $\theta \leftarrow \theta - \nabla_\theta \ell_{\text{denoiser}}(\theta, \mathcal{B}),$
        $\theta_{\text{EMA}} \leftarrow \text{EMA}_{0.9999}(\theta_{\text{EMA}}, \theta)$

---

**Sampling.** The walk-jump sampling approach is very flexible and allows us to configure sampling in different ways. For example, we can choose the number of walk steps between jumps, the maximum number of walk steps per chain or the number of chains run in parallel. Different sampling hyperparameters can change the statistics of samples, e.g., samples that are close to each other on a sample chain will likely be similar molecules. Therefore, we decided to fix some sampling hyperparameters for benchmarking purposes. In all our quantitative experiments, we generate samples in the following way: (i) we initialize all the chains $y_0$ in parallel, (ii) we "walk" $K$ steps with Langevin MCMC to sample smoothed codes $y_K$, and (iii) we "jump" with the denoiser (in a single step) to get the clean codes $\hat{z}_K$. In practice, we sampled 10000 molecules using 1000 MCMC steps for both QM9 and GEOM-drugs, and 10000 steps for CREMP on a single A100 GPU.

---

**Algorithm 4:** Sampling pseudo-code - the For loop corresponds to walk steps

**Input** $\delta$ (step size), $\gamma$ (friction), $K$ (steps), denoiser $\hat{z}_\theta$ trained at noise level $\sigma$.
$y_0 \sim \mathcal{U}_d(\min_{z \in \mathcal{D}_z, i \in \{1 \cdots d\}} z_i, \max_{z \in \mathcal{D}_z, i \in \{1 \cdots d\}} z_i) + \mathcal{N}(0, \sigma^2 I_d)$
$u_0 \leftarrow 0$
**for** $k = 0, \dots, K-1$ **do**
    $y_{k+1/2} \leftarrow y_k + \frac{\delta}{2} u_k$
    $g \leftarrow g_\theta(y_{k+1/2}) \triangleq (\hat{z}_\theta(y_{k+1/2}) - y_{k+1/2})/\sigma^2;$          /* score Equation (5) */
    $u_{k+1/2} \leftarrow u_k + \frac{\delta}{2} g$
    $u_{k+1} \leftarrow \exp(-\gamma\delta) u_{k+1/2} + \frac{\delta}{2} g + \sqrt{(1 - \exp(-2\gamma\delta))}\varepsilon, \quad \varepsilon \sim \mathcal{N}(0, I_d)$
    $y_{k+1} \leftarrow y_{k+1/2} + \frac{\delta}{2} u_{k+1}$
**Output** $\hat{z}_K \leftarrow \hat{z}_\theta(y_K)$ ;          /* jump step (denoising) */

---

**From codes to molecules.** After generating modulation codes, we need to extract the atom types and coordinates from them. This is a constrained optimization problem, and we provide a simple algorithm to find its solution: (i) render a voxel grid representation of the molecule at resolution of .25Å (tensors of dimensions $5 \times 32 \times 32 \times 32$, $8 \times 64 \times 64 \times 64$ and $6 \times 96 \times 96 \times 96$ on QM9, GEOM-drugs and CREMP, respectively), (ii) find the peaks of the voxel grids—they correspond to a discretized version of the atomic coordinates—with a simple $3 \times 3 \times 3$ kernel, and (iii) find the local optima of the atomic coordinates with the approach described in Section 3.3. Our continuous refinement approach leverages L-BFGS with learning rate 1.0 and is batched across 100 molecules of same size.

**Assets used in this work.** Our code is available at https://github.com/prescient-design/funcmol. Our neural field code is based on the open source implementation of MFN from [31] and the conditional version from [27]. Our code for walk-jump sampling is based on the open source implementation of VoxMol from [5]. Our metrics are computed using code from [54] and RDKit [88]. Our datasets *GEOM-drugs* [13], *CREMP* [12] and QM9 [81] are downloaded from the corresponding webpages. We use the protein visualization tool of [98]. All these assets are available publicly and to our knowledge have a CC-BY 4.0 license.

## C Analysis of the latent space

We perform three experiments to qualitatively explore the learned manifold and show empirically that it is well structured.

First, we pick several pairs of molecules and show the interpolation trajectory in latent modulation space. We project the interpolated codes back to the learned manifold of molecules via a noise/denoise operation. Figure 5 illustrates six trajectories, where we observe that molecules close in latent space share similar structure.

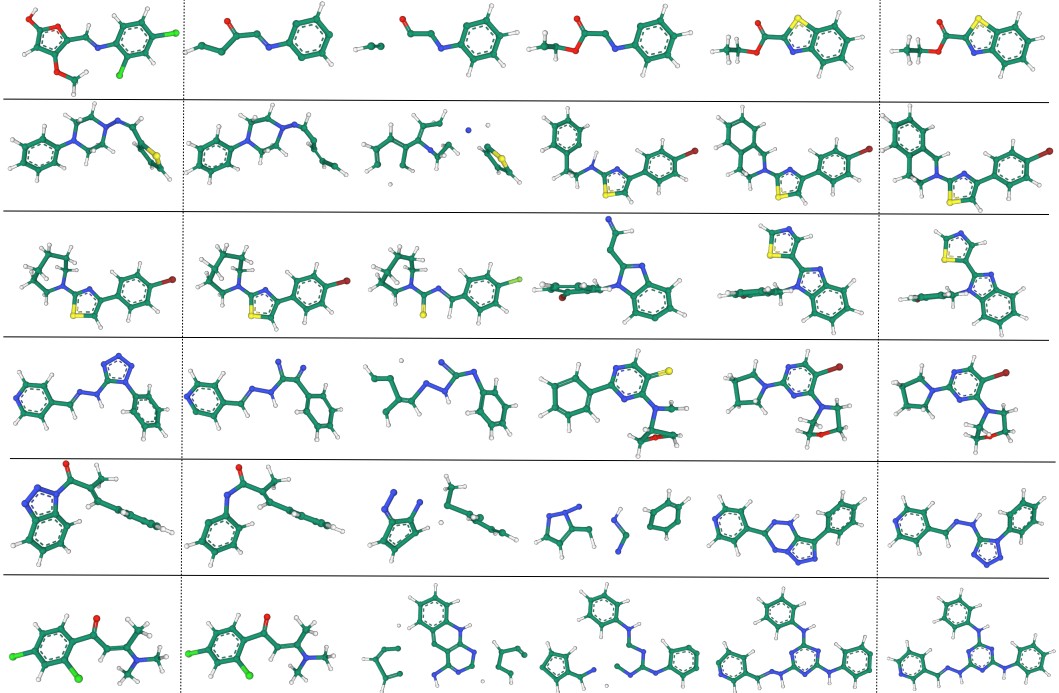

Figure 5: Interpolation in the latent modulation space for different pairs of molecules from GEOM-drugs. Each interpolated codes is protected back to the learned manifold of molecules via a noise/denoise operation. FuncMol produces semantically meaningful patterns in the interpolated space and we observe that molecules close in latent space share similar structure.

Second, we show t-SNE plots to demonstrate that the modulation space $z$ encodes molecular properties of QM9. For four different properties, we use t-SNE to embed 400 molecules divided equally between those with the highest and those with the lowest property values. Figure 6 shows that molecules with similar property values cluster together.

Finally, we evaluate the latent codes on downstream tasks. We train a linear regression model on frozen latent codes (a.k.a. linear probing) to see how the learned modulations correlate with different properties. Figure 7 shows the scatter plots and Spearman correlation for four different properties. We observe that the codes are highly predictive of the considered properties, despite being trained in an unsupervised fashion.

## D Diffusion baseline

We consider one additional model, $FuncMol_{dec, diff}$ for the auto-decoding setting. This model is similar to $FuncMol_{dec}$ but we sample codes with a diffusion model instead of walk-jump sampling. We use the same neural field and modulation codes as in $FuncMol_{dec}$ and we train a multi-level denoiser (with 1000 levels of noise) instead of a single-level one. The modulation codes are sampled like in standard diffusion models: we start from a Gaussian noise and iteratively apply the denoiser until we arrive on clean codes. We tried to train the diffusion variant of the model on GEOM-drugs,

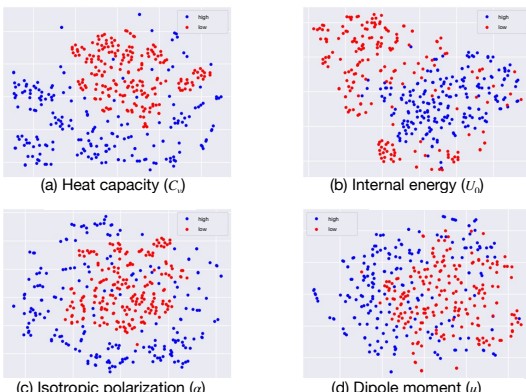

Figure 6: t-SNE plots of latent modulations codes of QM9 molecules for different molecular properties. For each plot, we pick 200 molecules from validation set with high value of a property (blue) and 200 with low value (red). We show results for four properties: (a) heat capacity ($C_v$), (b) internal energy ($U_0$), (c) isotropic polarization ($\alpha$) and (d) dipole moment ($\mu$).

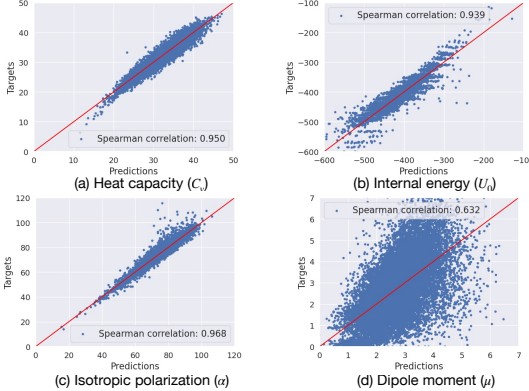

Figure 7: Performance of linear regression model (a.k.a linear probing) trained on modulation codes to predict molecular properties on QM9. We show the scatter plots and Spearman correlation for four different properties: (a) heat capacity ($C_v$), (b) internal energy ($U_0$), (c) isotropic polarization ($\alpha$) and (d) dipole moment ($\mu$).

but we were not successful (the generated molecules had very low stability). Table 4 shows the performance of the diffusion model relative to our other models. It performed worse than walk-jump sampling but was still able to generate good quality molecules.

Table 4: Comparing walk-jump sampling and diffusion model on QM9 results (10000 samples per model). ↑/↓ indicate that higher/lower numbers are better. The row data are randomly sampled molecules from the validation set.

| | stable mol %↑ | stable atom%↑ | valid %↑ | unique %↑ | valency $W_{1↓}$ | atom TV↓ | bond TV↓ | bond len $W_{1↓}$ | bond ang $W_{1↓}$ | time s/mol.↓ |
|---|---|---|---|---|---|---|---|---|---|---|
| FuncMol | 89.4 | 99.1 | 100. | 93.1 | .021 | .012 | .006 | .004 | 1.49 | 0.05 |
| FuncMol$_{\text{dec}}$ | 88.6 | 99.2 | 100. | 81.1 | .022 | .066 | .032 | .006 | 1.21 | 0.05 |
| FuncMol$_{\text{dec, diff}}$ | 70.8 | 97.3 | 95.8 | 81.1 | .007 | .034 | .021 | .006 | 1.25 | 0.07 |

# E  Ablations

## E.1  Reconstruction quality

We analyze the reconstruction quality of the fields and molecules we compressed with latent codes.

**Field reconstruction.**    We report in Table 5 the reconstruction performance of the molecular fields using our conditional MFN architecture and learned codes.

Table 5: Ablation: field reconstruction (whole training set).

| dset | MSE $\downarrow$ | PSNR $\uparrow$ |
|---|---|---|
| GEOM-drugs | $2.8 \cdot 10^{-6}$ | 55.5 |
| CREMP | $2.9 \cdot 10^{-6}$ | 55.4 |
| QM9 | $6.1 \cdot 10^{-6}$ | 52.1 |

**Molecule reconstruction.**    To make more sense out of these raw reconstruction metrics, we show that the 3D molecules decoded from learned training codes are valid and stable molecules. This means that we successfully compressed the training data into low-dimensional latent vectors via our field-based representation. Table 6 reports metrics of molecules decoded from the learned training codes for GEOM-drugs and QM9. For each dataset, we display the metrics of the training molecules, the molecules decoded from our training codes and the molecules derived from a voxelized representation of the field at a resolution of .25Å. We observe that the molecules rendered from codes have better metrics than those derived from voxels showing the validity of our new representation. These results are an upper bound to the results in Table 2.

Table 6: Ablation: molecule reconstruction (sample of 4k).

| dset | type | stable mol %$_\uparrow$ | stable atom%$_\uparrow$ | valid %$_\uparrow$ | valency $W_{1\downarrow}$ | atom TV$_\downarrow$ | bond TV$_\downarrow$ | bond len $W_{1\downarrow}$ | bond ang $W_{1\downarrow}$ |
|---|---|---|---|---|---|---|---|---|---|
| GEOM-drugs | *data* | 99.9 | 99.9 | 100. | .001 | .001 | .025 | .000 | .05 |
|  | code | 83.1 | 99.5 | 100. | .188 | .007 | .026 | .004 | .19 |
|  | voxel | 83.7 | 99.4 | 93.8 | .252 | .006 | .026 | .001 | .43 |
| QM9 | *data* | 98.7 | 99.8 | 98.9 | .001 | .003 | .000 | .000 | .12 |
|  | code | 95.5 | 99.7 | 100. | .010 | .009 | .003 | .001 | .16 |
|  | voxel | 92.5 | 99.4 | 98.8 | .017 | .009 | .002 | .002 | .30 |

## E.2  Atomic coordinate refinement

A big advantage of using neural fields is that we can represent signals, here molecules, in continuous space rather than in discrete space as in voxel representations. The continuous refinement introduced in Section 3.3 improves the quality of the molecules by finding more precise atomic coordinates. Table 7 shows the improvement of this continuous refinement over the refinement used in [5] that operates in discrete space: by going to continuous space, we overcome the limitations of discrete grids and substantially improve the stability of the molecules and the angle between bonds.

## E.3  Neural field robustness to noise

Here, we analyze how the neural field is robust to noise on the modulation code space. Figure 8 illustrates how molecular stability and the distance between the distribution of bond angles change as we increasingly add noise to the codes. Each point consists of the average of the metric over 4000 random modulation codes from the validation set. Interestingly, the neural field is quite robust to noise as we see that the metrics unchanged even at a reasonable amount of noise. We believe that this code robustness to noise helps better learn the denoiser.

Table 7: Ablation: continuous refinement improvement on code reconstruction performance. Metrics computed with 4000 generated samples on validation reference set.

| dset | FuncMol coord refine | stable mol %↑ | stable atom%↑ | valid %↑ | valency $W_{1\downarrow}$ | atom $TV_\downarrow$ | bond $TV_\downarrow$ | bond len $W_{1\downarrow}$ | bond ang $W_{1\downarrow}$ |
|---|---|---|---|---|---|---|---|---|---|
| GEOM- | ✓ | 83.1 | 99.5 | 100. | .188 | .007 | .026 | .004 | .189 |
| drugs | ✗ | 78.8 | 96.2 | 100. | .090 | .007 | .018 | .011 | 2.71 |
| QM9 | ✓ | 95.5 | 99.7 | 100. | .010 | .009 | .003 | .001 | .158 |
| | ✗ | 79.1 | 96.4 | 100. | .009 | .008 | .002 | .011 | 2.76 |

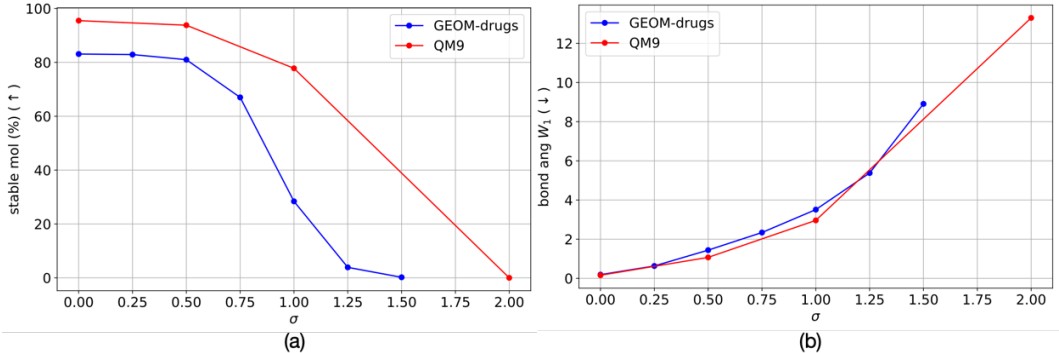

Figure 8: Ablation: code robustness to noise on GEOM-drugs (blue) and QM9 (red). Stable molecule (a) and bond angle distance (b) metrics as we increasingly add noise to the codes. Metrics are computed with 4000 generated samples on validation reference set.

### E.4 Number of walk steps

Table 8 shows how the quality of molecules changes as we increase the number of walk steps $K$ in the walk-jump sampling for FuncMol on GEOM-drugs. In this experiment, we sample 2000 samples and compute metrics w.r.t the validation set. First, we observe that some metrics improve (e.g., molecular stability) while other get worse (e.g., uniqueness). Second, we observe that the walk-jump chain is extremely stable, allowing us to perform as much as 50000 MCMC steps in the chain without breaking it. Finally, sampling time does not increase significantly: going from 500 to 50000 steps only results in a $10\times$ increase in sampling time (this is because the sampling bottleneck is on finding the atomic coordinates).

Table 8: Ablation on the number of walk steps $K$ on GEOM-drugs. Metrics computed with 2000 generated samples on test reference set.

| $K$ (n steps) | stable mol %↑ | stable %↑ | unique %↑ | valency $W_{1\downarrow}$ | atom $TV_\downarrow$ | bond $TV_\downarrow$ | bond len $W_{1\downarrow}$ | bond ang $W_{1\downarrow}$ | avg. t s/mol.↓ |
|---|---|---|---|---|---|---|---|---|---|
| 500 | 52.8 | 98.1 | 99.8 | .235 | .116 | .031 | .003 | 2.58 | .279 |
| 1000 | 68.8 | 98.8 | 97.4 | .246 | .109 | .051 | .003 | 2.51 | .298 |
| 2000 | 77.1 | 99.0 | 93.7 | .247 | .108 | .068 | .003 | 2.86 | .337 |
| 5000 | 80.6 | 99.0 | 85.6 | .247 | .150 | .091 | .003 | 3.37 | .456 |
| 10000 | 82.4 | 99.0 | 77.9 | .247 | .162 | .109 | .003 | 3.52 | .654 |
| 20000 | 83.8 | 98.9 | 73.6 | .252 | .154 | .130 | .004 | 3.52 | 1.05 |
| 50000 | 84.9 | 99.0 | 66.6 | .259 | .166 | .158 | .003 | 3.44 | 2.24 |

### E.5 Impact of resolution at decoding time

Finally, we measure the impact of resolution on sampling time and quality. As expected, sampling time becomes slower as we we have finer resolution. We also notice that finer resolution has better results than coarse ones (although the results stop improving). We chose resolution 0.25Å as it provides a good trade-off between performance and speed. The table below shows the results as a function of resolution.

Table 9: Ablation on the impact of resolution on sampling quality on GEOM-drugs. Metrics computed with 2000 generated samples on test reference set.

| resolution Å | stable mol %$_\uparrow$ | stable atom %$_\uparrow$ | valid %$_\uparrow$ | unique %$_\uparrow$ | valency $W_{1\downarrow}$ | atom $TV_\downarrow$ | bond $TV_\downarrow$ | bond len $W_{1\downarrow}$ | bond ang $W_{1\downarrow}$ | avg. t s/mol.$_\downarrow$ |
|---|---|---|---|---|---|---|---|---|---|---|
| 0.167 | 68.6 | 98.8 | 100. | 97.4 | .246 | .109 | .052 | .003 | 2.52 | .89 |
| 0.25 | 68.8 | 98.8 | 100. | 97.4 | .250 | .109 | .052 | .003 | 2.51 | .29 |
| 0.5 | 58.8 | 97.7 | 100. | 98.0 | .247 | .096 | .051 | .003 | 2.46 | .08 |

## F  Additional quantitative results

We report some additional plots for evaluation on GEOM-drugs. See Section 5 for more details.

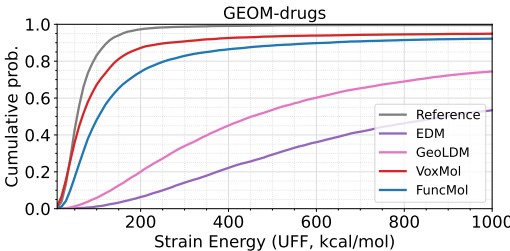

Figure 9: Cumulative distribution function of strain energy of generated molecules on GEOM-drugs based on 10000 molecules.

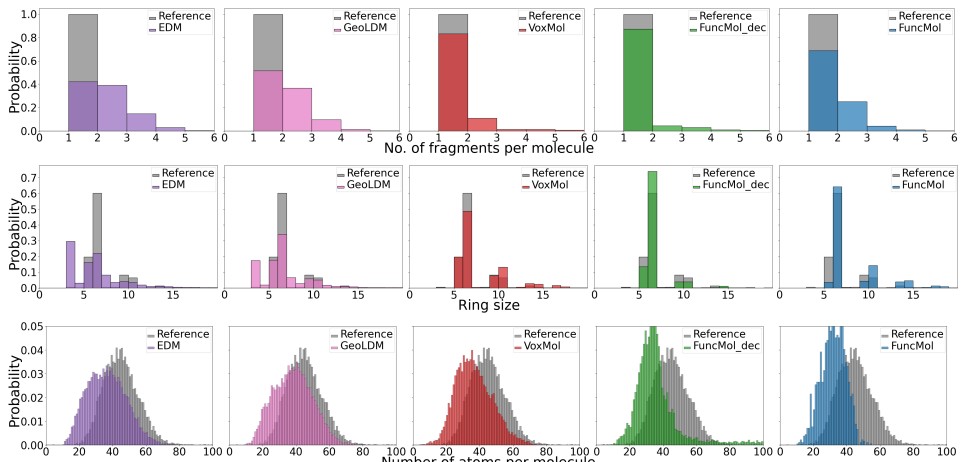

Figure 10: Histograms (over 10000 samples) showing (first row) distribution of number of fragments, (second) distribution of ring size, and (third) distribution of number of atoms per molecule.

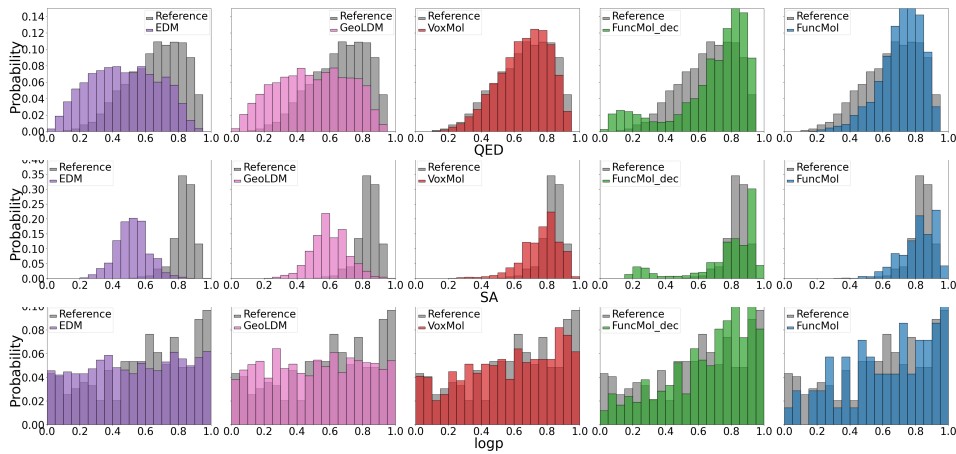

Figure 11: Histograms (over 10000 samples) showing (top) distribution of QED, (mid) SA and (bottom) log p.

## G  Comparison to bond-diffusion baselines

Some recent papers e.g., MolDiff [56], MiDi [54], show that incorporating extra information such as bonds and formal charges in point cloud-based approaches (e.g. EDM) improves the quality of the generated samples. These contributions are orthogonal to ours and are an interesting future addition to FuncMol, e.g. via additional channels in the molecular field.

We compare FuncMol to MolDiff despite different training assumptions, for completeness. MolDiff only incorporates bond information into the diffusion process, making it a simple representative baseline for this class of model. Since the weights for MolDiff with hydrogens were unavailable, we compared FuncMol using MolDiff's metrics and the MolDiff performance reported in their Appendix D.1, Table 8. The results are reported in Table 10. We observe that FuncMol achieves competitive results in most metrics despite not leveraging bond information.

Table 10: Comparison of MolDiff with H and FuncMol

|  | MolDiff with H | FuncMol |
|---|---|---|
| Validity ↑ | 0.957 | 1.000 |
| Connectivity ↑ | 0.772 | 0.739 |
| Succ. Rate ↑ | 0.739 | 0.739 |
| Novelty ↑ | 1.000 | 0.992 |
| Uniqueness ↑ | 1.000 | 0.977 |
| Diversity ↑ | 0.427 | 0.810 |
| Sim. Val. ↑ | 0.695 | 0.554 |
| QED ↑ | 0.688 | 0.715 |
| SA ↑ | 0.806 | 0.815 |
| Lipinski ↑ | 4.868 | 5.000 |
| RMSD ↓ | 1.032 | 1.088 |
| JS bond lengths ↓ | 0.414 | 0.529 |
| JS bond angles ↓ | 0.182 | 0.217 |
| JS dihedral angles ↓ | 0.244 | 0.232 |

# H    Additional qualitative results

We display some generated molecules in Figures 12 and 13 and display a MCMC chain in Figure 14.

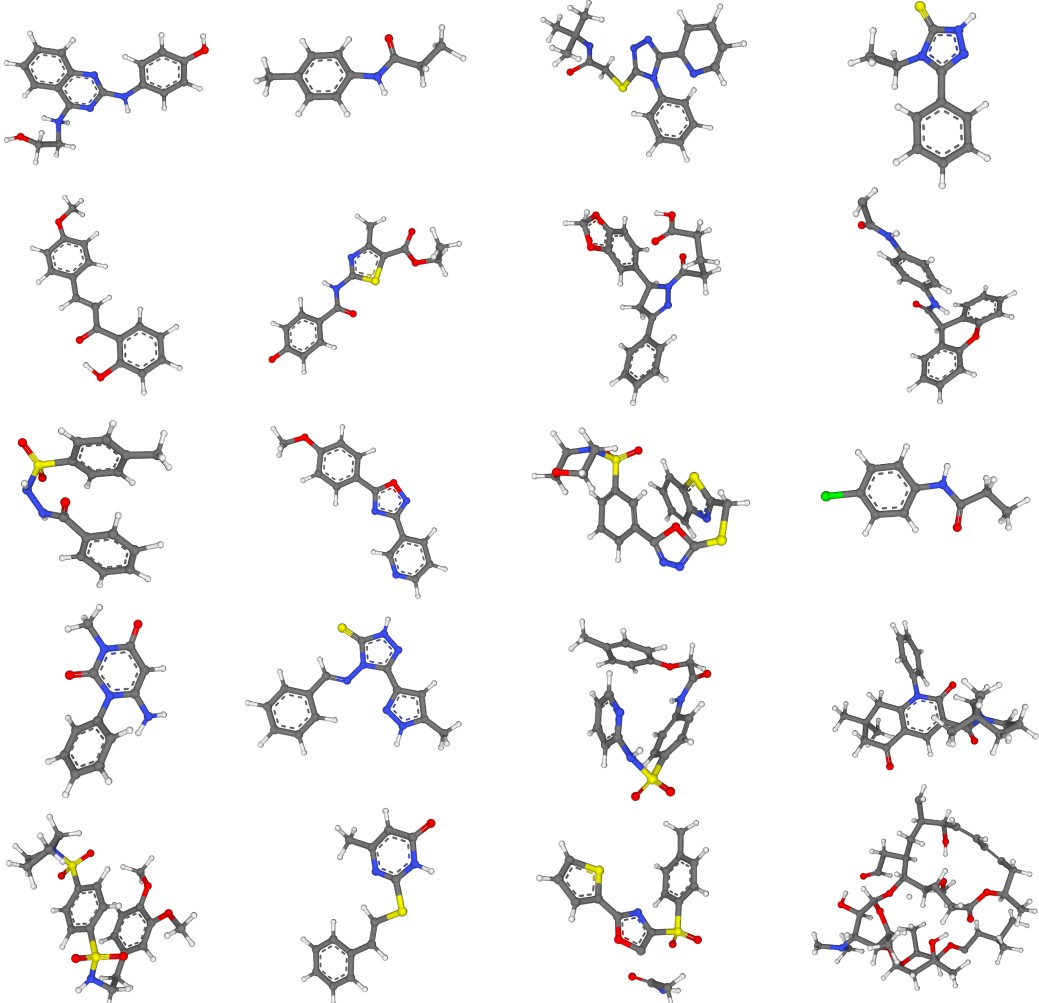

Figure 12: Generated samples from FuncMol trained on GEOM-drugs.

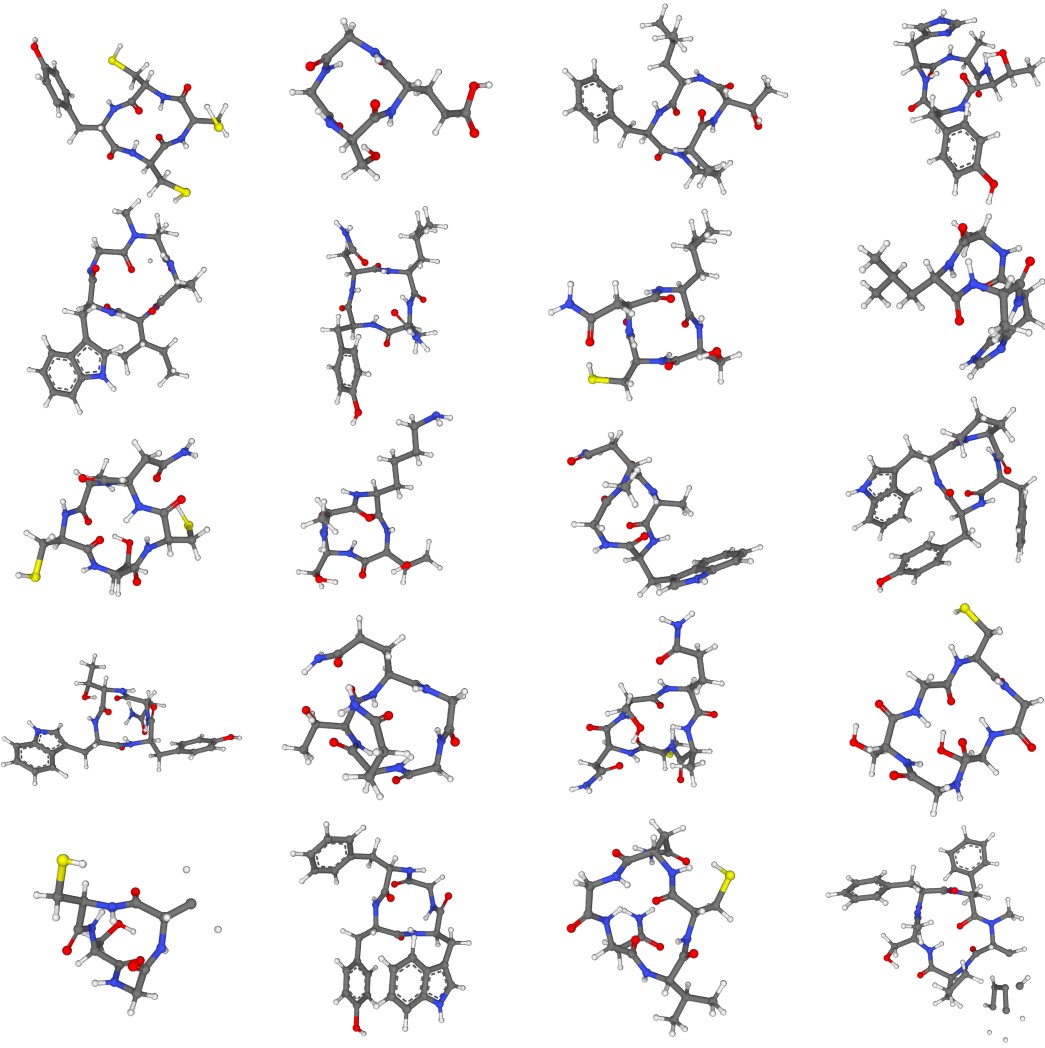

Figure 13: Generated samples from FuncMol trained on CREMP.

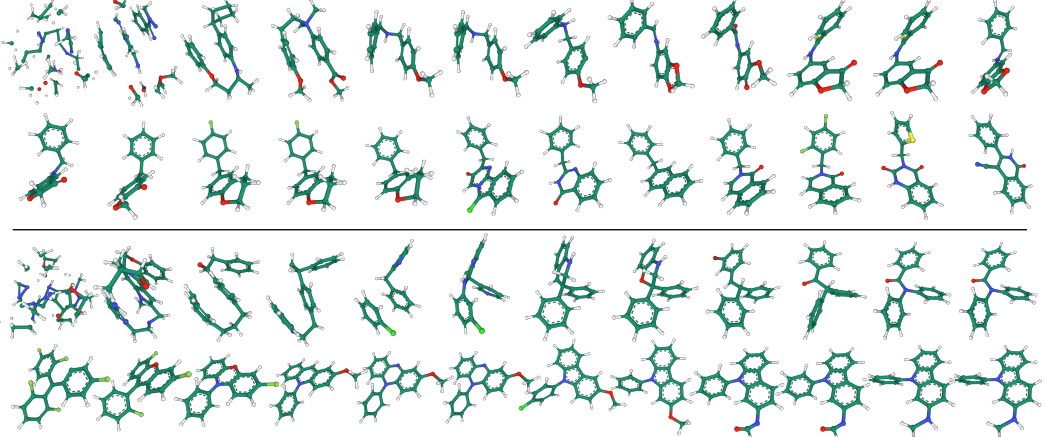

Figure 14: Two single MCMC chains generated by FuncMol, initialized randomly with different seeds (seen from left to right, top to bottom). Molecules are generated after each 200 "walk" steps.

