# OpenReview forum: "Score-based 3D molecule generation with neural fields"
_NeurIPS.cc/2024/Conference — NeurIPS 2024 poster_

### Official Review · Reviewer_NVMn · 2024-07-09

**Soundness:** 3
**Presentation:** 3
**Contribution:** 3
**Rating:** 6
**Confidence:** 4

**Summary:**

In this study the authors propose a novel FuncMol model for molecular generation. The proposed approach is based on continuous space neural fields and involves the joint training of a molecular neural field along with molecular modulation codes. The implementation of a neural empirical Bayes technique allows the generation of new molecular structures during the inference phase. The performance of the proposed model is compared against established neural baseline models on the GEOM-DRUGS dataset, which comprises small drug-like molecules. Additionally, the model is examined on the CREMP dataset, containing macrocyclic peptides, to provide a comprehensive analysis of its capabilities.

**Strengths:**

To the best of my knowledge, the idea of using continuous space neural fields for the generation of molecules is novel and previously unexplored. Moreover, in addition to training on the popular GEOM dataset, the authors also train the model on the CREMP dataset, which contributes to the advancement of peptide generation domain.

**Weaknesses:**

While the approach presented in the paper is novel in the context of molecular generation, the experimental analysis is incomplete. Several baselines and thorough ablation studies to justify the selection of hyperparameters are absent, as is an in-depth examination of the model's limitations. Further details can be found in the Questions section.

**Questions:**

1. (major) The authors have incorporated several diffusion models as baselines, such as EDM and GeoLDM, to benchmark their model's performance on the GEOM-DRUGS dataset. However, they have missed the important MolDiff (a) baseline, which mitigates the problems of generated spatial structures including the atom-bond inconsistency problem by introducing guidance from auxiliary bond predictors. Moreover, MolDiff enhances the evaluation framework by incorporating additional Jensen-Shannon divergence metrics to assess structural and spatial aspects more comprehensively.
2. (major) The section on macrocycle peptide generation does not take into account other relevant baselines, one of which is RINGER (b), whose authors released code for reproduction.
3. (moderate) The experimental setup in this study focuses only on unconditional molecule generation. In the practice, spatial molecular structures generation is mainly applied given some conditions, like conformation generation (c) and pocket-conditioned generation (d), which respectively aim to generate spatial molecular structures based on a given graph or a protein pocket. The paper would benefit from extending its analysis to demonstrate how the model could be adapted to integrate external conditions and discussing potential avenues for this adaptation.
4. (minor) The current model is not SE(3) equivariant, which implies that molecular rotations and translations could affect the modulation codes. The authors note this as a limitation in line 346 but do not specify whether any techniques were utilized, such as coordinate canonicalization, to mitigate this issue during model training. Further clarification on the methods used to address this challenge would be valuable.
5.(minor) One of the model bottlenecks is reconstructing atom positions from the continuous occupancy field, which necessitates calculations on a uniform discretization grid with cubic scaling. The authors select a discretization step of 0.25Å, followed by L-BFGS optimization for refining atom positions. However, the manuscript does not include an ablation study examining the impact of different discretization steps on the balance between model performance and computational speed, which would be critical for justifying the chosen discretization step.

(a) MolDiff: Addressing the Atom-Bond Inconsistency Problem in 3D Molecule Diffusion Generation

(b) RINGER: Rapid Conformer Generation for Macrocycles with Sequence-Conditioned Internal Coordinate Diffusion

(c) Torsional Diffusion for Molecular Conformer Generation

(d) 3D Equivariant Diffusion for Target-Aware Molecule Generation and Affinity Prediction

**Limitations:**

In the "Discussion" section, the authors address several model limitations; however, they do not explore the significant issue of integrating external conditions into the model.

---

> ### Author Rebuttal · Authors · 2024-08-05
>
> We thank the reviewer for the comments and questions. A general rebuttal is posted above. Below we address the reviewer's individual questions.
>
> **1. MolDiff Baseline.** MolDiff shows that incorporating bond information in point cloud-based approaches improves the quality of the generated samples. This contribution is orthogonal to ours and can potentially be incorporated into our generative model, e.g. via additional channels. This, however, is not the focus of our work, nor that of the baselines we considered. Here, we aim at proposing for the first time neural fields as a new representation for 3D molecules (a non-trivial task).
>
> For completeness (despite different training assumptions), we compare FuncMol to MolDiff. Since the weights for MolDiff with hydrogens were unavailable, we compared FuncMol using MolDiff’s metrics and the MolDiff performance reported in their Appendix D.1, Table 8. We observe that FuncMol achieves competitive results in most metrics despite not using bond information.
>
> |                | MolDiff with H | FuncMol |
> |----------------|---------------|----------------|
> | Validity ↑                   | 0.957        | 1.000     |
> | Connectivity ↑           | 0.772        | 0.739     |
> | Succ. Rate ↑             | 0.739        | 0.739     |
> | Novelty ↑                   | 1.000        | 0.992    |
> | Uniqueness ↑            | 1.000        | 0.977    |
> | Diversity ↑                 | 0.427        | 0.810    |
> | Sim. Val. ↑                 | 0.695        | 0.554    |
> | QED ↑                       | 0.688        | 0.715    |
> | SA ↑                          | 0.806        | 0.815    |
> | Lipinski ↑                   | 4.868        | 5.000   |
> | RMSD ↓                    | 1.032        | 1.088    |
> | JS bond lengths ↓     | 0.414        | 0.529    |
> | JS bond angles ↓      | 0.182        | 0.217    |
> | JS dihedral angles ↓ | 0.244        | 0.232    |
>
> **2. RINGER Baseline.** To our knowledge, we are the first to report results on CREMP in the unconditional all-atom 3D molecule generation setting. This experiment is more for qualitative purposes as CREMP is very recent and not a standard benchmark for this task. RINGER [1] also considered CREMP but tackled conformer generation, a different problem from ours: it assumes knowledge of the molecular sequence/graph during training and sampling. This simplifies generation, since the model knows a priori the number of atoms, their types, the bonds between them and the approximate atom locations. RINGER only parametrizes angles and torsions, while FuncMol and its baselines perform all-atom generation.
> We tried to extend our baselines to CREMP but did not succeed, mainly due to the high memory consumption (e.g. VoxMol took 40 GPU-hours per epoch, while FuncMol took 2.7 GPU-hours). To encourage comparisons to FuncMol, we include some quantitative metrics used in [1] that measure the distance between test and generated distributions using KL divergence. The KL divergence for bond angles are 0.1615 ($\theta_1$), 0.1345 ($\theta_2$) and 0.2197 ($\theta_3$); the ones for the dihedral angles are 0.1127 ($\phi$), 0.1178 ($\psi$) and 0.1813 ($\omega$). The percentage of valid generated peptides (for which we can extract the sequence of amino acids from their SMILES) is 82.7%.
>
> **3. Focus on unconditional generation, which is not as good as conditional generation.** We agree that conditional generation presents more practical value than unconditional generation. We note, however, that our approach is the first to introduce neural fields as a representation of 3D molecules. The focus of our submission is to show that (i) it is a feasible approach (this is non-trivial), (ii) it scales well with data and molecule size, and (iii) it achieves competitive results with fast sampling time. We will include a discussion about this topic on the appendix and we will leave extensions to conditional generation for future work (similar to how TargetDiff [1]/DiffSBDD [2] extend EDM or VoxBind [3] extends VoxMol).
>
> **4. The model is not SE(3) equivariant. How to leverage rotation/translation?** We do not leverage any SE(3) equivariance on the approach described in the original manuscript. We acknowledge that this is a good point and address it in the rebuttal by performing data augmentation during training (see main Rebuttal comment "Issues with memorization"). Data augmentation improves the quality of the representations as it helps learn a "more semantic" latent space z. This is reflected in the empirical results of Rebuttal Tables 1 and 2, where the uniqueness score significantly improves.
>
> **5. Ablation study missing: Impact of different discretization steps.** Please, see the main Rebuttal, where we address this comment. Appendix D of the manuscript contains four other thorough ablation studies that justify other design choices in FuncMol.
>
> [1] Guan et al. "3D Equivariant Diffusion for Target-Aware Molecule Generation and Affinity Prediction". ICLR23
>
> [2] Schneuing et al. "Structure-based Drug Design with Equivariant Diffusion Models". Arxiv22
>
> [3] Pinheiro et al. "Structure-based drug design by denoising voxel grids" ICML24

---

> ### Comment · Reviewer_NVMn · 2024-08-12
>
> I would like to thank the authors for their answers on my inquiries, especially concerning the ablation study and the comparative analysis with the MolDiff model. I hope, that all provided metrics and the new baselines will be included in the final version. Additionally, it would be beneficial to include results from the MolDiff model trained on a hydrogen-depleted graph. I raise my score.

---

### Official Review · Reviewer_eB4P · 2024-07-11

**Soundness:** 3
**Presentation:** 3
**Contribution:** 2
**Rating:** 7
**Confidence:** 3

**Summary:**

This paper proposes a new representation for 3d molecules, and the representations are low-dimensional, compact, and scalable. Based on the representation, this paper proposes a new score-based generative model, FuncMol, which shows competitive reuslts on GEOM-drug and scales up to CREMP. Besides, FuncMol adopts the recently proposed Walk-Jump sampling and enjoys fast sampling.

**Strengths:**

1. New representation. This paper proposes atom occupancy fields as Gaussian-like 3D shapes, enabling "density" estimation for any points in the space.
2. Feasible decoding. Decoding a molecule (atom types and coordinates) directly from an embedding vector is extremely difficult, yet decoding a molecular field from an embedding vector seems more feasible.
3. Competitive results. Based on the proposed molecular neural field, this paper proposes a generative model and achieves competitive results with SOTA models, making this representation potential in downstream applications.
4. Fast sampling.

**Weaknesses:**

1. From the perspective of 3d molecule generation, FuncMol doesn't outperform VoxMol and GeoLDM in almost all metrics. It could be a trade-off between generative ability and representation quality.
2. The authors claim FuncMol can scale up to larger molecules, yet the more fundamental problem is, can we have a scaling law for molecules like LLM. Scaling law is essential for molecular foundation models, and should have a much larger impact.
3. Molecule generation may not be a perfect scenario for this representation, which seems more like a generative pretraining framework. Besides, the reviewer is interested in the application of such representations on downstream applications, which are not tested, nor discussed.

I will raise my score if the authors present more evidence for weakness 3.

**Questions:**

1. Why FuncMol cannot outperform GeoLDM and VoxMol in many metrics?
2. Why use walk-jump sampling? It seems unnecessary to "jump" before the last step.
3. How to guarantee the generalization of learned "codes"? The manifold for valid molecules may not be continuous, or support the embedding space. Thus, the reviewer suspects that random initialization and Langevin MCMC may produce invalid molecules.

I will raise my score if the questions are properly answered.

**Limitations:**

See above

---

> ### Author Rebuttal · Authors · 2024-08-05
>
> We thank the reviewer for the comments and positive feedback. A general rebuttal is posted above. Below we address the specific issues raised by the reviewer.
>
> **“Why does FuncMol not outperform GeoLDM and VoxMol?"** It is challenging to explain why these models perform differently, as they make different choices w.r.t. data representation, NN architectures and generative modeling. FuncMol outperforms GeoLDM in most of the metrics of Rebuttal Table 2 and some metrics of Rebuttal Table 1 (c.f. main rebuttal). VoxMol outperforms FuncMol on GEOM-drugs (the performance gap is smaller when adding data augmentation to FuncMol). VoxMol uses 3D UNets, an expressive and well-established NN architecture tailored to discrete data, while FuncMol uses relatively new neural field architectures; neural fields tend to underperform against discrete-data architectures in many computer vision applications [1]. FuncMol, however, generates 3D molecules an order of magnitude faster than VoxMol and GeoLDM. Moreover, unlike them, FuncMol easily scales to larger molecules (such as macro-cyclic peptides).
>
> **Scaling laws for molecules.** We agree that coming up with scaling laws for 3D molecules is an interesting research direction. However, this lies beyond the scope of this work, especially when taking into account the large amount of computation resources required for this type of study. That being said, we observed that at fixed dataset, increasing the parameter count of our models (both the neural field and our denoiser) significantly improved our results. Moreover, increasing the dataset e.g. with data augmentation leads to better performance.
>
> **Downstream applications: "Molecule generation may not be a perfect scenario for this representation".**  We thank the reviewer for this comment that helped improve the paper. Generative modeling for molecules is an important task with many practical applications. It also serves as a testbed to demonstrate that neural fields provide a very expressive/efficient/scalable decoder for 3D molecules, unlike point clouds or voxel grids.
>
> At the reviewer’s request, we show that neural field representations can also be used in other downstream tasks, e.g. discriminative tasks related to property prediction. We consider some properties from QM9 ($\mu$ dipole moment, $u_0$ internal energy, $\alpha$ isotropic polarizability, $C_v$ heat capacity) and report the Spearman correlation performance (ranging from -1 to 1) in the linear probing setting (i.e. training a linear regression on top of frozen codes). We observe high ranking scores and correlations between predictions and ground truth labels as shown on Rebuttal Fig 4 (c.f. main rebuttal pdf) and the table below.
>
> |                | $\mu$ | $u_0$ | $\alpha$ | $C_v$ |
> |----------------|---------------|----------------|----------|-----------|
> | Spearman_rho  ↑       | 0.632          | 0.939           | 0.968     | 0.950      |
>
> **Why use walk-jump sampling?** We explain why we use walk-jump sampling/neural empirical Bayes on L40-41: "[it] enjoys many properties such as fast-mixing, simplicity for training and fast sampling speed" as it relies on a single noise level. It has been successfully applied in 3D molecule generation (e.g., VoxMol) and antibody sequence generation [2]. Despite choosing walk-jump sampling for our main results, we stress that our framework is agnostic to the generative model used; we reported results with diffusion models on Appendix C. Compared to diffusion, we found that walk-jump sampling on this task is simpler/faster to train, faster to sample from (less sampling steps are required) and achieves slightly better empirical results.
>
> **How to guarantee the generalization of learned "codes"? Random initialization and Langevin MCMC may produce invalid molecules.** We agree with the reviewer that the manifold of valid molecules is very complicated (low-dimensional in a large ambient space and potentially non-smooth). However, the manifold of codes after _gaussian-smoothing_ is much simpler, especially with large noise. This allows us to perform Langevin MCMC from random initialization. This is precisely the intuition of neural Empirical Bayes [3], the generative framework we used in this work. We observed empirically that Langevin MCMC on smoothed codes mixes well. For example, Rebuttal Fig. 5 (c.f. main rebuttal pdf) shows two single MCMC chains initialized randomly with different seeds, where molecules are generated after each 200 "walk" steps. This phenomenon has also been observed on different data modalities, e.g., images [4], biological sequences [2] and voxelized molecules (VoxMol).
>
> [1] Dupont et al. "From data to functa: Your data point is a function and you can treat it like one." ICML2022.
>
> [2] Frey et al. "Protein Discovery with Discrete Walk-Jump Sampling". ICLR24
>
> [3] Saremi et al.. "Neural empirical Bayes". JMLR19
>
> [4] Saremi et al.. “Multi-measurement Generative Models”. ICLR22

---

> > ### Comment · Reviewer_eB4P · 2024-08-14
> >
> > I raised my score

---

### Official Review · Reviewer_R2Qo · 2024-07-11

**Soundness:** 3
**Presentation:** 3
**Contribution:** 3
**Rating:** 7
**Confidence:** 4

**Summary:**

This paper introduces FuncMol, a method that leverages recent work on implicit neural representations for unconditional 3D molecule generation. The key idea is to (1) parametrize continuous atomic densities through a neural field network and molecule-specific modulation codes and (2) use a score-based generative model in modulation code space to efficiently generate new molecules. The authors demonstrate competitive performance on the GEOM-drugs dataset, showing that the generated molecules are roughly on par with existing techniques at significantly faster sampling speeds.

**Strengths:**

* Using implicit neural representations to represent 3D point clouds and atomic densities is an innovative approach to 3D molecular generation that could complement the current graph- and voxel-based state-of-the-art.
* The score-based generative model and walk-jump sampling process in latent space afford significantly faster sampling speeds than existing approaches and enable the method to scale favorably to larger systems.
* The paper is well-written and the authors clearly describe their method and the motivation behind it, as well as the experimental details used for training and evaluation.

**Weaknesses:**

* The approach seems to have some issues with overfitting/memorization, which the authors mitigate by applying dropout between the MFNs multiplicative blocks. However, the model still seems to generate significantly fewer unique molecules than the baselines. It would be good if the authors could comment on whether this is caused by e.g. limited capacity of the neural field, a degenerate modulation code space, or the walk-jump sampling scheme, and if there are any ways to address this.
* The method performs strictly worse than existing approaches on the QM9 dataset. The authors argue that QM9 is not a suitable dataset to evaluate unconditional generative models, since it consists of an exhaustively enumerated set of small molecules. However [2] and [3] only state that novelty is not a meaningful performance metric in this setting. The ability to capture this relatively well-behaved training distribution and generate stable and unique molecules should still be a reasonable test of generative modeling performance.
* The paper only provides qualitative results for the CREMP dataset. Since the scalability of the proposed method to larger systems is listed as one of its main advantages, it would be helpful to include a quantitative comparison to other methods from [1]. Most of the metrics in Table 1 should be directly applicable to macro-cyclic peptides, in addition to e.g. the percentage of closed rings and canonical side chains. Similarly, it would be good to include the sampling times in this setting.

**Questions:**

* What is the rationale behind switching from the meta-learning approach in [4] to optimizing the modulation codes and neural field parameters jointly? Could this cause some of the issues with generating novel/unique molecules?
* Tables 1 and 2 report metrics for 10000 samples per model. However, in lines 257-263, the authors mention that the model without dropout needs to generate twice as many samples as the model with dropout. Are the models generating more than 10000 samples and are non-novel molecules filtered out before reporting the results? If not, what is the percentage of novel molecules for each model?
* [2] defines *atom stability* as the proportion of atoms that have the correct valency and *molecule stability* as the proportion of generated molecules for which all atoms are stable. How can the percentage of valid molecules be higher than the percentage of stable molecules, since RDKit’s sanitization filter checks if a molecule has correct valencies?

---

[1] Grambow, Colin A., et al. "RINGER: Rapid Conformer Generation for Macrocycles with Sequence-Conditioned Internal Coordinate Diffusion." arXiv preprint arXiv:2305.19800 (2023).

[2] Hoogeboom, Emiel, et al. "Equivariant diffusion for molecule generation in 3d." International conference on machine learning. PMLR, 2022.

[3] Vignac, Clement, and Pascal Frossard. "Top-N: Equivariant Set and Graph Generation without Exchangeability." International Conference on Learning Representations.

[4] Dupont, Emilien, et al. "From data to functa: Your data point is a function and you can treat it like one." International Conference on Machine Learning. PMLR, 2022.

**Limitations:**

The authors provide a brief discussion of the lack of equivariance as a limitation in Section 6.

---

> ### Author Rebuttal · Authors · 2024-08-05
>
> We thank the reviewer for the valuable feedback. A general rebuttal is posted above. Below we address the reviewer's specific questions.
>
> **Issues with overfitting/memorization.** By further investigating this issue, we realized that memorization is likely caused by a degenerate latent space—as alluded to by the reviewer. Data augmentation helped mitigate this issue and learn a more "semantically meaningful" latent space. This is reflected in the results of Rebuttal Table 1 and 2 (see main Rebuttal "Issues with memorization"), where uniqueness is significantly improved and we no longer observe memorization. This is a consequence of the model overfitting less, which also allowed us to train denoisers with higher capacity.
>
> **Worse performance on QM9.**  We agree that QM9 is a good task to test generative models. FuncMol and VoxMol perform comparably on QM9 and both are outperformed by point cloud approaches. FuncMol, however, has an order of magnitude faster sampling time. QM9 has limited relevance for drug discovery: the number of stable and unique molecules is limited as QM9 is an enumeration of all molecules up to 9 heavy atoms satisfying some constraint. Achieving a high performance on QM9 does not mean the model can handle more complex distributions. In fact, the best models we reported on QM9 (GeoLDM, EDM) perform the worst on GEOM-drugs.
>
> **Quantitative results on CREMP.** To our knowledge, we are the first to report results on CREMP in the unconditional all-atom 3D molecule generation setting. This experiment is more for qualitative purposes as CREMP is very recent and not a standard benchmark for this task. RINGER [1] also considered CREMP but tackled conformer generation, a different problem from ours: it assumes knowledge of the molecular sequence/graph during training and sampling. This simplifies generation, since the model knows a priori the number of atoms, their types, the bonds between them and the approximate atom locations. RINGER only parametrizes angles and torsions, while FuncMol and the baselines perform all-atom generation.
> We tried to extend our baselines to CREMP but did not succeed, mainly due to the high memory consumption (e.g. VoxMol took 40 GPU-hours per epoch, while FuncMol took 2.7 GPU-hours). To encourage comparisons to FuncMol, we include some quantitative metrics used in [1] that measure the distance between test and generated distributions using KL divergence. The KL divergence for bond angles are 0.1615 ($\theta_1$), 0.1345 ($\theta_2$) and 0.2197 ($\theta_3$); the ones for the dihedral angles are 0.1127 ($\phi$), 0.1178 ($\psi$) and 0.1813 ($\omega$). The percentage of valid generated peptides (for which we can extract the sequence of amino acids from their SMILES) is 82.7%.
>
> **Sampling time on CREMP.** We reported the sampling time on L327: "our model takes around 1.4s to generate a molecule. [..] should VoxMol be trained successfully, it would take over a minute [per] molecule".
>
> **Why switch from the meta-learning approach in [6]?** Joint optimization, a.k.a. auto-decoding has been applied in several works [2, 5] to learn signed-distance functions. Auto-decoding is attractive as it does not require the memory-expensive double loop optimizations in meta-learning and is simpler to optimize. We attribute the novelty/uniqueness issue to the lack of data augmentation during training (see "Issues with memorization" in the main Rebuttal).
>
> **Filtering samples due to memorization.**  As written on L261: "we only consider the novel molecules of FuncMol$ _{drop=0}$ and exclude those pathological chains when benchmarking this model"; the total number of molecules that are evaluated is 10,000. The other reported FuncMol model (with dropout) considers both novel and non novel molecules. Our initial novelty scores are 56.3\% for FuncMol$ _{drop=0}$ and 77.3\% for FuncMol (with dropout). With data augmentation, the novelty score is over 99\%.
>
> **How can the percentage of valid molecules be higher than the percentage of stable molecules?** We use the same metrics as previous work for an apples to apples comparison to other work. In fact, all baselines that report these two metrics have higher validity than mol stability (EDM, GeoLDM, VoxMol). _Validity_ is measured as the success rate of RDKit sanitization and is computed only on the largest fragment of the generated molecules (when they consist of disjoint fragments). _Atom_ and _molecule stability_ are defined as the reviewer described and were introduced in [4]. As explained by [3]: "[mol stability] is similar to validity but, in contrast to RDKit sanitization, they do not allow for adding implicit hydrogens to satisfy the valency constraints."
>
> [1] Grambow et al. "RINGER" Arxiv23
>
> [2] Park et al. “DeepSDF” CVPR19
>
> [3] Vignac et al. "MiDi" ECML23
>
> [4] Satorras et al. "E(n) equivariant normalizing flows" NeurIPS21
>
> [5] Chou et al. "Diffusion-SDF" ICCV23
>
> [6] Dupont et al. "From data to functa" ICML22

---

> > ### Comment · Reviewer_R2Qo · 2024-08-12
> >
> > I would like to thank the authors for the detailed response. The proposed improvements and reported performance gains strengthen the empirical section of the paper and address my main concerns. I will raise my score accordingly.

---

### Official Review · Reviewer_yg8b · 2024-07-12

**Soundness:** 4
**Presentation:** 4
**Contribution:** 3
**Rating:** 7
**Confidence:** 4

**Summary:**

This study proposes a neural field model that treats molecular data as continuous atomic occupancy fields. The model learns a latent code that can be used to predict the atomic occupancy in discretized grids. The authors then perform score-based generative modeling using neural empirical Bayes and show higher generation quality than point cloud-based and voxel-based models.

**Strengths:**

This study utilizes a novel representation of 3D molecular structures. The proposed model does not require pre-defined atom numbers, which is a common limitation of point cloud representations, has higher expressivity than GNN-based methods, and also scales better than voxel grid representations. It could also scale to larger molecules such as cyclic peptides.

**Weaknesses:**

I only have some minor concerns and questions. See the "Questions" section.

**Questions:**

1\. It would be interesting to see whether the latent space has any meaningful manifold. For example, does the latent code capture patterns such as basic molecular fragments or similarity between molecules, etc.?

2\. Also, though the sampling is fast, the peak finding and iterative refinement could be time-consuming. Could the authors comment on this?

3\. Some recent methods [1,2] explicitly model bonds, leading to high quality generations. How does the proposed method compare with these methods? Is it also possible to incorporate bond information into the molecular field modeling, e.g. as additional channels?

[1] Peng, Xingang, et al. "Moldiff: Addressing the atom-bond inconsistency problem in 3d molecule diffusion generation." arXiv preprint arXiv:2305.07508 (2023).

[2] Vignac, Clement, et al. "Midi: Mixed graph and 3d denoising diffusion for molecule generation." Joint European Conference on Machine Learning and Knowledge Discovery in Databases. Cham: Springer Nature Switzerland, 2023.

**Limitations:**

The authors have properly addressed the limitations.

---

> ### Author Rebuttal · Authors · 2024-08-05
>
> We thank the reviewer for the positive and valuable feedback. A general rebuttal is posted above. Below we address the reviewer's concerns.
>
> **1. Meaningful representation of latent space.** See "Meaningful representation of latent space [yg8b]" on the main Rebuttal. Overall, the additional experiments show that the latent space is well structured.
>
> **2. Peak finding + iterative refinement can be time consuming.** To clarify, the average sampling time reported on Table 2 is the time required to fully generate the molecules ("from noise to set of atom types/coordinates"). It includes four steps: (i) the walk-jump sampling steps to sample latent codes, (ii) rendering (going from the code to the voxel grid at resolution .25A), (iii) the peak finding and (iv) the iterative refinement. The sampling time for all steps together is an order of magnitude faster than alternative baselines.
> The bottleneck in sampling time is the rendering phase (ii). The sampling time can be reduced at the cost of performance by rendering codes at a lower resolution (see "Ablation study" on the main Rebuttal).
>
> **3. Explicitly modeling bonds between atoms.** Some recent works e.g., MolDiff, MiDi, show that incorporating extra information such as bonds and formal charges in point cloud-based approaches (e.g. EDM) improves the quality of the generated samples. These contributions are orthogonal to ours and can potentially be incorporated into our generative model, e.g. via additional channels as suggested by the reviewer. This is, however, not the focus of our work (nor that of the baselines we considered). Here, we aim at proposing for the first time neural fields as a new representation for 3D molecules (a non-trivial task).
>
> For completeness (despite different training assumptions), we compare FuncMol to MolDiff. MolDiff only incorporates bond information into the diffusion process, making it a simple representative baseline for this class of model. Since the weights for MolDiff with hydrogens were unavailable, we compared FuncMol using MolDiff’s metrics and the MolDiff performance reported in their Appendix D.1, Table 8. We observe that FuncMol achieves competitive results in most metrics despite not leveraging bond information during training.
>
> | Metric               | MolDiff with H | FuncMol |
> |----------------|---------------|----------------|
> | Validity ↑                   | 0.957        | 1.000     |
> | Connectivity ↑           | 0.772        | 0.739     |
> | Succ. Rate ↑             | 0.739        | 0.739     |
> | Novelty ↑                   | 1.000        | 0.992    |
> | Uniqueness ↑            | 1.000        | 0.977    |
> | Diversity ↑                 | 0.427        | 0.810    |
> | Sim. Val. ↑                 | 0.695        | 0.554    |
> | QED ↑                       | 0.688        | 0.715    |
> | SA ↑                          | 0.806        | 0.815    |
> | Lipinski ↑                   | 4.868        | 5.000   |
> | RMSD ↓                    | 1.032        | 1.088    |
> | JS bond lengths ↓     | 0.414        | 0.529    |
> | JS bond angles ↓      | 0.182        | 0.217    |
> | JS dihedral angles ↓ | 0.244        | 0.232    |

---

> > ### Comment · Reviewer_yg8b · 2024-08-12
> >
> > I thank the authors for addressing my concerns and questions.

---

### Author Rebuttal · Authors · 2024-08-05

We thank the reviewers for the helpful comments. The reviewers agree that the paper has good soundness, presentation and contributions. They also acknowledge that our approach is novel, addresses limitations of point-cloud and voxel representations, is scalable, has faster sampling and is competitive.

Next, we highlight the new experiments of the rebuttal (*see Rebuttal Figures in the attached pdf and Rebuttal Tables in the responses*). We address each review in their respective section.

**Issues with memorization [R2Qo]. How to leverage data augmentation [NVMn]?** We trained a new model with data augmentation. Instead of using *auto-decoding* [1] to learn the modulation codes (c.f. manuscript), we use an *auto-encoding* approach [2],  which allows us to leverage data augmentation on the fly. In this setting, z is computed with a (trainable) 3D CNN encoder that takes as input a voxelized molecule (Rebuttal pdf Fig 1). Once the encoder is learned, we use it to generate samples z with random data augmentation and train a denoiser as described in Section 4.1. This approach leads to a more structured latent space z as reflected in Rebuttal Table 1 and 2. We no longer observe memorization and uniqueness is significantly improved.

**Rebuttal Table 1:**
| GEOM-drugs         | stable mol %↑ | stable atom %↑ | valid %↑ | unique %↑ | valency W₁↓ | atom TV↓ | bond TV↓ | bond len W₁↓ | bond ang W₁↓ |
|----------------|---------------|----------------|----------|-----------|--------------|----------|----------|---------------|--------------|
| **data**                                            | 99.9 | 99.9 | 99.8 | 100.0  | .001 | .001 | .025 | .000 | 0.05 |
| **EDM**                                           | 40.3 | 97.8 | 87.8 | 99.9 | .285 | .212 | .048 | .002 | 6.42 |
| **GeoLDM**                                    | 57.9 | 98.7 | 100. | 100. | .197 | .099 | .024 | .009 | 2.96 |
| **VoxMol**                                       | 75.0 | 98.1 | 93.4 | 99.6 | .254 | .033 | .024 | .002 | 0.64 |
| **FuncMol$ _{autodec}$**               | 60.6 | 98.2 | 100. | 86.6 | .244 | .079 | .044 | .003 | 2.05 |
| **FuncMol$_{autodec,drop}$**       | 69.7 | 95.3 | 100.  | 77.5 | .268 | .035 | .028 | .003 | 2.13 |
| **FuncMol$ _{autoenc}$**               | 72.6 | 99.1 | 100. | 95.6 | .250 | .107 | .046 | .003 | 2.31 |

**Rebuttal Table 2:**
| GEOM-drugs  | single frag %↑ | median energy↓ | ring sz TV↓ | atms/mol TV↓ | QED ↑ | SA ↑ | logp ↑ | time s/mol.↓ |
|------------------|----------------|----------------|-------------|--------------|-------|------|--------|--------------|
| **data**                                       | 100.0 | 54.5  | .011 | .000 | .658 | .832 | 2.95 | -    |
| **EDM**                                      | 42.2  | 951.3 | .976 | .604 | .472 | .514 | 1.11 | 9.35 |
| **GeoLDM**                               | 51.6  | 461.5 | .644 | .469 | .497 | .593 | 1.05 | 8.96 |
| **VoxMol**                                 | 82.6  | 69.2  | .264 | .636 | .659 | .762 | 2.73 | 7.55 |
| **FuncMol$ _{autodec}$**         | 86.0 | 105.0   | .263 | .677 | .713 | .862 | 2.87 | 0.69  |
| **FuncMol$ _{autodec,drop}$** | 80.2 | 96.4 | .324 | .970 | .677 | .788 | 2.87 | 0.53  |
| **FuncMol$ _{autoenc}$**         | 73.9  | 111.7 | .434 | .878 | .715 | .816 | 3.03 | 0.86 |

**Ablation study missing: impact of different discretization steps [NVMn].** As suggested by the reviewer, we include a new ablation study that measures the impact of resolution on sampling time and quality. We observe that finer resolutions improve performance but also increase sampling time. We chose resolution 0.25A as it provides a good trade-off between performance and speed. The table below shows the results as a function of resolution:

| resolution     | stable mol %↑ | stable atom %↑ | valid %↑ | unique %↑ | valency W₁↓ | atom TV↓ | bond TV↓ | bond len W₁↓ | bond ang W₁↓ | time per mol (s)↓ |
|----------------|---------------|----------------|----------|-----------|--------------|----------|----------|---------------|--------------|--------------|
| **0.167A** | 72.8 | 99.1 | 100. | 95.8 | .248 | .127 | .042 | .003 | 2.36 | 2.8 |
| **0.25A**   | 72.6 | 99.1 | 100. | 95.6 | .250 | .127 | .046 | .003 | 2.41 | .86 |
| **0.5A**     | 51.7 | 97.6 | 100. | 96.2 | .246 | .126 | .046 | .003 | 2.29 | .15 |

For reference, we had already included four ablation studies in Appendix D.4.

**Meaningful representation of latent space [yg8b]; downstream task and “generalization of learned codes” [eB4P].**  We perform three experiments to qualitatively explore the learned manifold and show empirically that it is well structured.
* First, we pick several pairs of molecules and show the interpolation trajectory in latent modulation space. We project the interpolated codes back to the learned manifold of molecules via a noise/denoise operation. Rebuttal pdf Fig 2 illustrates six trajectories, where we observe that molecules close in latent space share similar structure.
* Second, we show t-SNE plots to demonstrate that the modulation space z encodes molecular properties of QM9. For four different properties, we use t-SNE to embed 400 molecules divided equally between those with the highest and those with the lowest property values. Rebuttal pdf Fig 3 shows that molecules with similar property values cluster together.
* Finally, we evaluate the latent codes on downstream tasks. We train a linear regression model on frozen latent codes (a.k.a. linear probing) to see how the learned modulations correlate with different properties. Rebuttal pdf Fig 4 shows the scatter plots and Spearman correlation for four different properties. We observe that the codes are highly predictive of the considered properties, despite being trained in an unsupervised fashion.

[1] Park et al. "DeepSDF Learning continuous signed distance functions for shape representation"CVPR19

[2] Mescheder et al. "Occupancy networks Learning 3d reconstruction in function space"CVPR19

---

### Decision · Program_Chairs · 2024-09-25

**Decision:**

Accept (poster)

**Comment:**

A breath of fresh air for molecular generation, implemented and presented well. The key idea is to learn an atom position density instead of learning atomic positions explicitly (or some roto-translational invariant featurization), using an implicit neural representation. The authors parameterize the atomic densities to depend on a 'modulation code.' At inference time, new molecules are sampled from a smoothed distribution of the modulation code using a combination of Langevin MCMC and walk-jump sampling. A gap remains when compared to particle-based methods in terms of sampling time. No reviewers have highlighted this limitation, which might be due to their limited experience within this domain. However, the authors show potential for scaling to larger systems, as illustrated here using macrocycles.

Nevertheless, the model generates a density as its output, suggesting that an inverse problem needs to be solved post hoc to get a molecular structure, not unlike fitting of experimental data such as CryoEM densities. This fact may prove to be a significant computational overhead for larger systems.